



# Representing inter-annual land cover and vegetation variability based on satellite observations in the HTESSEL land surface model

Fransje van Oorschot[1,2], Ruud J. van der Ent[2], Markus Hrachowitz[2], Emanuele Di Carlo[1], Franco Catalano[3], Souhail Boussetta[4], Gianpaolo Balsamo[4], and Andrea Alessandri[1]

[1]Institute of Atmospheric Sciences and Climate, National Research Council of Italy (CNR-ISAC), Bologna, Italy
[2]Department of Water Management, Faculty of Civil Engineering and Geosciences, Delft University of Technology, Delft, The Netherlands
[3]Italian National Agency for New Technologies, Energy and Sustainable Economic Development (ENEA), Rome, Italy
[4]European Centre for Medium-Range Weather Forecasts (ECMWF), Reading, UK

**Correspondence:** Fransje van Oorschot (f.vanoorschot@tudelft.nl)

**Abstract.** Vegetation largely controls land surface-atmosphere interactions. Although vegetation is highly dynamic across spatial and temporal scales, most land surface models currently used for reanalyses and near-term climate predictions do not adequately represent these dynamics. This causes deficiencies in the variability of modeled water and energy states and fluxes from the land surface. In this study we evaluated the effects of integrating spatially and temporally varying land cover and vegetation characteristics derived from satellite observations on modelled evaporation and soil moisture in the Hydrology Tiled ECMWF Scheme for Surface Exchanges over Land (HTESSEL) land surface model. Specifically, we integrated inter-annually varying land cover from the European Space Agency Climate Change Initiative, and inter-annually varying Leaf Area Index (LAI) from the Copernicus Global Land Services (CGLS). Additionally, satellite data of the Fraction of green vegetation Cover (FCover) from CGLS was used to formulate and integrate a spatially and temporally varying effective vegetation cover parameterization. The effects of these three implementations on model evaporation fluxes and soil moisture were analysed using historical offline (land-only) model experiments at the global scale, and model performances were quantified with global observational products of evaporation ($E$) and near-surface soil moisture ($\mathrm{SM_s}$). The inter-annually varying land cover consistently altered the evaporation and soil moisture in regions with major land-cover changes. The inter-annually varying LAI considerably improved the correlation of $\mathrm{SM_s}$ and $E$ with respect to the reference data, with largest improvements in semiarid regions with predominantly low vegetation during the dry season. These improvements are related to the activation of soil moisture-evaporation feedbacks during vegetation-water-stressed periods with inter-annually varying LAI in combination with inter-annually varying effective vegetation cover, defined as an exponential function of LAI. The further improved effective vegetation cover parameterization consistently reduced the errors of model effective vegetation cover, and it regionally improved $\mathrm{SM_s}$ and $E$. Overall, our study demonstrated that the enhanced vegetation variability consistently improved the near-surface soil moisture and evaporation variability, but the availability of reliable global observational data remains a limitation for complete understanding of the model response. To further explain the improvements found, we developed an interpretation framework for how the model development activates feedbacks between soil moisture, vegetation, and evaporation during vegetation-water-stress periods.



## 1 Introduction

Land surface-atmosphere interactions are largely controlled by vegetation, which is dynamic across spatial (local, regional and global) and temporal (seasonal, inter-annual and decadal) scales (Seneviratne et al., 2010). Land surface models (LSMs) aim to describe these interactions and are, therefore, a crucial aspect of models used for climate reanalysis and climate predictions. However, most state-of-the-art LSMs do not adequately represent the temporal and spatial variability of vegetation, resulting in weaknesses in the associated variability of modelled surface water and energy states and fluxes (e.g. Alessandri et al., 2007;

Pitman et al., 2009; Ukkola et al., 2016; Fisher and Koven, 2020; Hersbach et al., 2020; Van Oorschot et al., 2021).

Satellite remote sensing based data products provide observations that can be used in land surface models. Global satellite derived maps of land cover, leaf area index, or albedo have been directly used as boundary conditions in such models (Faroux et al., 2013; Alessandri et al., 2017; Boussetta et al., 2021). Furthermore, satellite products have been used to improve model parameterizations of e.g. surface roughness, soil characteristics, and subsurface water storage (Lo et al., 2010; Trigo et al.,

2015; Yang et al., 2016; Orth et al., 2017). Moreover, LSMs have been evaluated using global satellite products of e.g. land surface temperatures, snow depth, and soil moisture (Balsamo et al., 2018; Johannsen et al., 2019; Dong et al., 2020; Nogueira et al., 2020, 2021; Boussetta et al., 2021).

Recent studies have exploited latest satellite campaigns to update land cover (LC) and Leaf Area Index (LAI) representation into the land surface model 'Carbon-Hydrology ECMWF Tiled Scheme for Surface Exchanges over Land' (CHTESSEL)

(Johannsen et al., 2019; Nogueira et al., 2020, 2021; Boussetta et al., 2021) as part of the Integrated Forecasting System (IFS) of the European Centre for Medium-range Weather Forecasts (ECMWF). These studies replaced the original fixed map of land cover from the Global Land Cover Characteristics (GLCC) dataset representing the early 1990s (Loveland et al., 2000) with an updated map obtained from the latest generation estimates of land cover from the European Space Agency Climate Change Initiative (ESA-CCI) (Poulter et al., 2015). Similarly, the LAI climatology from the Moderate Resolution Imaging

Spectroradiometer (MODIS) (Boussetta et al., 2013) was replaced with updated climatology from the recent Copernicus Global Land Service (CGLS) LAI dataset (Verger et al., 2014). The integration of these satellite-derived variables considerably reduced the bias of model land surface temperatures (Johannsen et al., 2019; Nogueira et al., 2020, 2021). In addition, Boussetta et al. (2021) showed an overall reduction of model annual mean evaporation bias when using the updated LC and LAI in CHTESSEL.

LAI in LSMs can be coupled to the effective vegetation cover ($C_{\text{eff}}$), which characterizes the density of the vegetated surface

from a top view that effectively contributes to the water and energy balances. The organization structure of leaves inside the canopy is reported as vegetation clumping. In previous modelling studies, the seasonal variations in $C_{\text{eff}}$ have been described as an exponential function of LAI considering vegetation clumping in (C)HTESSEL (Alessandri et al., 2017; Nogueira et al., 2020; Boussetta et al., 2021) and in other land modelling efforts (Anderson et al., 2005; Krinner et al., 2005; Le Moigne, 2012). The shape of the exponential relation between $C_{\text{eff}}$ and LAI in state-of-the-art land surface models has, to our knowledge been

assumed constant in time and space so far (Krinner et al., 2005; Alessandri et al., 2017; Nogueira et al., 2020; Boussetta et al.,



2021). However, studies have shown that the degree of vegetation clumping, and so the shape of this relation, actually varies for different vegetation types (Chen et al., 2005; Ryu et al., 2010; Zhang et al., 2014).

The research gap that we identified is that most previous LSM studies aimed at improving the temporally fixed boundary condition of land cover and the monthly seasonal cycle of LAI, while not exploring the effects of inter-annual variations of LC

and LAI. Moreover, previous studies have generally used one spatially fixed relationship between effective vegetation cover and LAI, while there is considerable evidence that this relationship is vegetation type dependent.

The objective of this research is to evaluate the effects of integrating temporal and spatial variations of land cover and vegetation characteristics derived from satellite observations on modelled evaporation and soil moisture in the land surface model HTESSEL. Specifically, we will integrate annually varying LC from ESA-CCI as well as seasonally and inter-annually

varying LAI from CGLS. Additionally, CGLS 'Fraction of green vegetation cover' (FCover; Verger et al., 2014) is used to formulate and implement a spatially (i.e. vegetation dependent) and temporally (i.e. inter-annually) variable effective vegetation cover parameterization in HTESSEL.

## 2 Methods

### 2.1 Earth observation products

Here we used yearly land cover maps at a $300\,\mathrm{m}$ spatial resolution from ESA-CCI for the time period 1993-2019 (Defourny et al., 2017; Copernicus Climate Change Service, 2019). In this dataset the land cover is classified into 22 classes based on the United Nations Land Cover Classification System (LCCS) (Di Gregorio and Jansen, 2005).

LAI and FCover data were obtained from CGLS for 1999-2019 (Copernicus Global Land Service, 2022). We used the 1km version 2 collection in which both products were derived at a 10-daily resolution from the top of canopy reflectance

measurements by the SPOT/VEGETATION (1999-2013) and PROBA-V (2014-2019) sensors (Verger et al., 2019). These two timeseries were homogenized using a cumulative distribution function (CDF) approach following Boussetta and Balsamo (2021). For model spin-up purposes, the CGLS LAI (1999-2019) was further extended backwards with former-generation LAI data from the Advanced Very-High Resolution Radiometer (AVHRR) for 1993-1999 at a 0.05° resolution (Pacholczyk and Verger, 2020). The AVHRR LAI (1993-1999) was interpolated using conservative interpolation (Schulzweida, 2022) to the

CGLS 1km resolution and harmonised with CGLS (1999-2019) using CDF-matching (Boussetta and Balsamo, 2021).

### 2.2 Model description

Here we used the HTESSEL land surface model (Balsamo et al., 2009) as it was developed and implemented for climate predictions with the EC-Earth3 Earth system model (Döscher et al., 2022). This version already implements a temporally, but not spatially, varying effective vegetation cover, which is further developed in this work (Alessandri et al., 2017). This section

describes the relevant model representations and adaptations.





### 2.2.1 Land cover representation

In HTESSEL the vegetated area of a grid cell is divided into high and low vegetation tiles. In case of snow there are separate model tiles for snow on bare ground/low vegetation and snow beneath high vegetation (Balsamo et al., 2009). Figure 1a represents an example of the vegetation types and cover fractions for a single grid cell, that were originally based on the GLCC
land cover dataset (Loveland et al., 2000). The low ($L$) and high ($H$) vegetation types with the largest cover fraction in each grid cell (see example in Fig. 1a) are used in HTESSEL as dominant vegetation types $T_L$ and $T_H$ (Fig. 1b). The corresponding HTESSEL vegetation cover fractions $A_L$ and $A_H$ are based on the total low and high vegetation grid cell cover fractions.

$T_L$ and $T_H$ directly control surface water and energy fluxes because model parameters such as vegetation root distribution, minimum canopy resistance, and roughness lengths for momentum and heat are obtained from lookup tables based on the
vegetation type (ECMWF, 2015). Surface fluxes are calculated separately for low and high vegetation tiles, and combined based on the fractions $A_L$ and $A_H$. Here we only focus on the surface evaporation flux, that we define as the sum of transpiration, soil evaporation, interception evaporation, and, in case of lakes, also open water evaporation (Savenije, 2004; Miralles et al., 2020). The subsurface in HTESSEL consists of 4 soil layers with depths of 7, 21, 100 and 189 cm. In this study we differentiate between near-surface soil moisture ($SM_s$) in the top layer (0-7 cm), and the subsurface soil moisture ($SM_{sb}$) in the three deeper
layers (7-289 cm).

### 2.2.2 Leaf Area Index representation

LAI is defined separately for the high and low vegetation tiles ($LAI_L$ and $LAI_H$). In the original HTESSEL model, $LAI_L$ and $LAI_H$ are prescribed as a seasonal cycle that is derived from a satellite-based climatology based on MODIS (Boussetta et al., 2013), and the vegetation cover fractions $A_L$ and $A_H$. The LAI controls the canopy resistance $r_c$ of the high and low vegetation
tiles through the following linear relation:

$$r_c = \frac{r_{s,min}}{LAI} f_1(R_s) f_2(D_a) f_3(\overline{SM}) \tag{1}$$

with $r_{s,min}$ the vegetation specific minimum canopy resistance and $f_1(R_s)$, $f_2(D_a)$ and $f_3(\overline{SM})$ functions describing the dependencies on shortwave radiation ($R_s$), atmospheric water vapor deficit ($D_a$), and weighted average soil moisture based on the root distribution over the four soil layers ($\overline{SM}$), respectively. The root fractions are generally the largest in soil layers 2
and 3, and, therefore, transpiration mostly origins from the $SM_{sb}$. The transpiration is linearly related to $r_c$ and atmospheric variables. Furthermore, the LAI controls the capacity of the model interception reservoir $W_{1m}$ by:

$$W_{1m} = W_{1max} * (C_B + C_L * LAI_L + C_H * LAI_H) \tag{2}$$

with $W_1max$=0.0002 m and $C_B$, $C_L$ and $C_H$ the fractions of bare soil, effective low and high vegetation, respectively (Sect. 2.2.3). The interception evaporation per time step follows from the water content of the interception reservoir (calculated from
precipitation), $W_{1m}$, and the potential evaporation.



### 2.2.3 Effective vegetation cover representation

The model effective low and high vegetation cover ($C_{\text{eff,L}}$ and $C_{\text{eff,H}}$) represent the part of the model vegetation cover fraction ($A_{\text{L}}$ and $A_{\text{H}}$) that is actively contributing to the water balance through transpiration and interception evaporation (Fig. 1c). The fraction of the grid cell not covered by effective vegetation is treated as bare soil ($C_{\text{B}}$), where only soil evaporation takes place.

Soil evaporation only occurs in the top soil layer (0–7 cm), and, therefore, origins only from $\text{SM}_{\text{s}}$. The model resistance to soil evaporation ($r_{\text{soil}}$) is described by

$$r_{\text{soil}} = r_{\text{soil,min}} f_3(\text{SM}_{\text{s}}) \tag{3}$$

with $r_{\text{soil,min}} = 50\,\text{sm}^{-1}$ and $f_3(\text{SM}_{\text{s}})$ representing the dependency on the first layer soil moisture content. The effective vegetation cover fractions $C_{\text{eff,L}}$ and $C_{\text{eff,H}}$ and bare soil fraction $C_{\text{B}}$, are described by:

$$C_{\text{eff,L}} = c_{\text{v,L}} * A_{\text{L}} \tag{4}$$

$$C_{\text{eff,H}} = c_{\text{v,H}} * A_{\text{H}} \tag{5}$$

$$C_{\text{eff}} = C_{\text{eff,L}} + C_{\text{eff,H}} \tag{6}$$

$$C_{\text{B}} = 1 - C_{\text{eff}} \tag{7}$$

with $c_{\text{v,L}}$ and $c_{\text{v,H}}$ representing the low and high vegetation density. Originally, $c_{\text{v,L}}$ and $c_{\text{v,H}}$ were described by a lookup table

with vegetation specific values, allowing for spatial variation of the $C_{\text{eff,L}}$, $C_{\text{eff,H}}$, and $C_{\text{B}}$ fractions. However, this approach does not represent temporal variations in vegetation density. To allow for temporal variability in $C_{\text{eff}}$ (represented by the arrows in Fig. 1c), $c_{\text{v,L}}$ and $c_{\text{v,H}}$ were linked to the seasonal variability of LAI by the following exponential relation (Alessandri et al., 2017):

$$c_{\text{v,L}} = 1 - e^{-k\,\text{LAI}_{\text{L}}} \tag{8}$$

$$c_{\text{v,H}} = 1 - e^{-k\,\text{LAI}_{\text{H}}} \tag{9}$$

with $k$ the canopy light extinction coefficient that represents the degree of vegetation clumping (Anderson et al., 2005). Previously $k$ was in general for all vegetation types set to constant values of 0.5 (Krinner et al., 2005; Alessandri et al., 2017) or 0.6 (Boussetta et al., 2021). As a consequence, the vegetation-dependent spatial variability in $k$ was not accounted for.

## 2.3 The implemented vegetation variability

### 2.3.1 Land cover variability

Here we implemented the annually varying ESA-CCI land cover (LC) data for the 1993-2019 period (Sect. 2.1), as developed by Boussetta and Balsamo (2021) for the HTESSEL vegetation types and spatial resolution. For consistency with the other model adaptations and evaluations (Sect. 2.3.2, 2.3.3 and 2.5), our LC analyses were based on 1999-2018. The inter-annually varying LC from ESA-CCI introduced a change in $T_{\text{L}}$ in 5%, and $T_{\text{H}}$ in 4% of the land grid cells between the first (1999)





and the last (2018) year of the considered study period (Fig. 2). Figure 2c shows the fraction of land grid cells in which each vegetation type (dominant in 1999) is replaced by another type in 2018 (plain colors), and conversely how often each vegetation type replaces the 1999 dominant one in 2018 (hatched colors). Crops and short grass relatively often replaced other low vegetation types, and evergreen needleleaf (EN) and deciduous broadleaf (DB) trees were relatively often replaced by other high vegetation types.

The low and high vegetation cover fractions changed in many regions according to the ESA-CCI LC dataset (Fig. 3). During the 1999-2018 period, low vegetation replaced high vegetation in the Southern Amazon and North-Eastern Siberia. Conversely, high vegetation replaced low vegetation in the boreal regions of Lapland and North-Western Siberia. Moreover, arid regions such as Central Asia and Australia experienced an expansion of low vegetation over the 1999-2018 period. In Fig. 3 we highlighted the Southern Amazon, Lapland and Central Asia regions where the vegetation cover fraction changed considerably. These regions are further analyzed in Sect. 3.1.

### 2.3.2 Leaf Area Index variability

We used the monthly CGLS LAI data described in Sect. 2.1 to prescribe model LAI, representing both the seasonal cycle and inter-annual variability of LAI. The $1\,\mathrm{km}$ LAI data was interpolated using conservative interpolation to the HTESSEL grid (Schulzweida, 2022). Next, LAI was disaggregated into low and high LAI ($\mathrm{LAI_L}$ and $\mathrm{LAI_H}$) based on the low and high vegetation cover fractions ($A_\mathrm{L}$ and $A_\mathrm{H}$), for the use in the HTESSEL model setup with separate low and high vegetation tiles (Boussetta et al., 2021; Boussetta and Balsamo, 2021). Figure 4 shows the LAI inter-annual variability as integrated here in HTESSEL, quantified with the standard deviation.

### 2.3.3 Vegetation specific effective vegetation cover parameterization

The CGLS FCover and LAI data were used (Sect. 2.1) to further develop the model effective vegetation cover parameterization as described by Eqs. (4)-(9). The constant $k$=0.5 parameter was replaced with a vegetation specific $k$ to improve spatial and temporal variability of the model $C_\mathrm{eff}$. We assumed that the model $C_\mathrm{eff}$ is equivalent to the CGLS FCover data. Following the model $C_\mathrm{eff}$ parameterization, FCover is then described as follows:

$$\mathrm{FCover} = 1 - e^{-k\,\mathrm{LAI}} \tag{10}$$

We estimated $k$ for different HTESSEL vegetation types by solving the minimization problem in Eq. (11) using a non-linear least squares optimization at a $1\,\mathrm{km}$ spatial resolution.

$$\min\|1 - e^{-k\,\mathrm{LAI}} - \mathrm{FCover}\|_2 \tag{11}$$

To discriminate vegetation types, the grid cells where each vegetation type maximizes its cover fraction based on the ESA-CCI LC developed in Boussetta and Balsamo (2021) were selected for each year. For each set of grid cells corresponding to each vegetation type, the FCover and LAI 10-daily, $1\,\mathrm{km}$ data for 1999-2019 were extracted. For the optimization of $k$, a randomly selected subsample of 2000 grid points of the LAI and FCover timesteps (10-daily) for each vegetation type was used to keep





the analysis computationally feasible, while ensuring a representative sample with robust significance of the fit. In this way, we obtained a sample of 2000 grid cells times 36 timesteps per year times 20 years, which equals 1 440 000 data points to be used for the optimization for each vegetation type. This optimization resulted in vegetation specific $k$-values that were implemented in the HTESSEL code as in Eqs. (4)-(9). The robustness of the optimization was verified by repeating the random selection
procedure several times, which resulted in negligible changes in the $k$-estimates.

## 2.4    Model experiments

We performed experiments with an offline, uncoupled version of HTESSEL to evaluate the effect of the implemented vegetation variability as described in Sect. 2.3. HTESSEL was forced with atmospheric hourly forcing from ECMWF Reanalysis v5 (ERA5) and simulations were performed from 1993-2019, with 1993-1999 as spin-up period (details in Table S1). The model
spatial resolution is the n128 reduced Gaussian grid corresponding to grid-cells of $\sim$75x75 km. In total, four different model experiments were performed (Table 1). In the first experiment, as a benchmark and control experiment (CTR) the land cover of all years was set to the ESA-CCI land cover of 1993, the LAI of all years was set to the 1993-2019 climatology and the $C_{\mathrm{eff}}$ parameterization with $k$=0.5 was used. This reflects standard settings of the EC-Earth3 version of HTESSEL. In the second experiment (IALC) the inter-annually varying ESA-CCI LC was included, while in the third experiment (IAK5) we further
added inter-annually varying CGLS LAI. Finally, the model sensitivity to the vegetation specific $C_{\mathrm{eff}}$ parameterization (see Sect. 2.2.3) was evaluated in the fourth experiment (IAKV). The model experiments were evaluated for 1999-2018, which is the longest period to consistently assess all three model implementations, with the available evaluation data described in Sect. 2.5.

## 2.5    Model evaluation

### 2.5.1    Model variables

The effects of the vegetation specific $C_{\mathrm{eff}}$ parameterization on the model $C_{\mathrm{eff}}$ were assessed in IAKV compared to IAK5. Furthermore, we analysed the effects of the increasingly detailed model land cover and vegetation variability in the three experiments (IALC, IAK5, IAKV) on total evaporation ($E$), and the evaporation components, i.e. transpiration ($E_{\mathrm{t}}$), soil evaporation ($E_{\mathrm{s}}$) and interception evaporation ($E_{\mathrm{i}}$). In addition, the effects on model near-surface soil moisture ($\mathrm{SM_s}$) and
subsurface soil moisture ($\mathrm{SM_{sb}}$) were analysed.

### 2.5.2    Reference data

The modeled $C_{\mathrm{eff}}$ was compared to the CGLS FCover data (Sect. 2.1) at the model spatial resolution. As a benchmark for total evaporation we used the 'Derived Optimal Linear Combination Evapotranspiration' version 3 (DOLCEv3), which is a linear combination of estimates from ERA5-land, GLEAM v3.5a and v3.5b and FLUXCOM-RSMETEO that was regionally
weighted based on the performance in reproducing FLUXNET tower evaporation observations (Hobeichi et al., 2021). The associated uncertainty estimate is spatially and temporally varying based on the gridded evaporation and flux tower observa-



tions (Hobeichi et al., 2018). This dataset was selected because it is intended to better capture evaporation temporal variations compared to previous DOLCE versions (v1 and v2) and was, therefore, found suitable for evaluating the effects of the modified temporal and spatial variability of vegetation on evaporation (Hobeichi et al., 2021). Daily evaporation and associated uncer-
tainty at a 0.25° resolution was used for 1999-2018 and was here interpolated using conservative interpolation (Schulzweida, 2022) to the model spatial resolution. Model near-surface soil moisture ($SM_s$) (0-7 cm) was compared to the ESA-CCI soil moisture product (ESA-CCI SM), that is based on multiple satellites with active and passive sensors (Dorigo et al., 2017; Gruber et al., 2019). This dataset provides near-surface ($\sim$0-5 cm) soil moisture at a daily resolution and a 0.25° grid. Here we used the combined active-passive product interpolated using conservative interpolation (Schulzweida, 2022) to the model spatial res-
olution for 1999-2018 (European Space Agency, 2022). The uncertainty estimates for ESA-CCI SM were also considered as they were provided with the data product and based on error variance of the data used to generate the product (Dorigo et al., 2017). ESA-CCI SM contains spatial and temporal gaps due to densely vegetated areas (tropical forests) and snow coverage. Here only grid cells with a temporal coverage larger than 60% were used, and, as a consequence, model performance metrics for $SM_s$ were only calculated for these grid cells.

### 2.5.3 Evaluation metrics

The hourly model output was first averaged to monthly values, based on which then annual means, monthly climatology, and inter-annual anomalies were calculated. To differentiate the seasons (June, July and August: JJA; September, October and November: SON; December, January and February: DJF; March, April and May: MAM), the monthly values were averaged to seasonal means, and inter-annual seasonal anomalies were calculated. For the evaluation of $E$ and $SM_s$ with respect to
reference data, we used the Pearson correlation coefficients $r$ of the inter-annual monthly and seasonal anomalies. To calculate $r$ of the inter-annual monthly and seasonal anomalies, the anomalies were detrended assuming a linear trend. Detrending was not applied for the effects of the modified LC, as the inter-annually varying LC mostly influenced the trend. In addition, we quantified the effects of the improved vegetation variability with the Root Mean Squared Error (RMSE). For $C_{eff}$ and $E$ RMSE we used monthly values, while for $SM_s$ we used standardized inter-annual anomalies. Model $SM_s$ and reference ESA-
CCI SM cannot be compared directly in absolute terms due to the different representative soil layers, and as a consequence, different temporal variability (Sect. 2.5.2). To overcome this limitation, we standardized the inter-annual anomalies for model and reference $SM_s$ by dividing the monthly $SM_s$ with the climatological standard deviation.

To test the significance of the $r$ and RMSE differences between models we used a bootstrap, in which 1000 data samples were randomly created by resampling the data of model 1 and model 2 with replacement for each timestep. We tested the null
hypothesis that the $r$ or RMSE of model 1 and model 2 with respect to the reference data are not significantly different from each other at the 10% significance level.





# 3 Results

## 3.1 Land cover inter-annual variability effects

The inter-annually varying land cover from ESA-CCI in experiment IALC resulted in a shift in mean evaporation components
(i.e. $E_t$, $E_s$, and $E_i$) compared to the CTR experiment (Fig. 5). The last five years of the simulations (2014-2019) are considered, because the effects of the inter-annually varying land cover mostly emerge in this period. In the Southern Amazon, where $A_H$ on average reduced from 0.64 to 0.57 in IALC compared to CTR (Fig. 3), the mean $E_t$ reduced by 3% from 633 to 615 $\mathrm{mm\,year^{-1}}$ and $E_i$ reduced by 6% from 384 to 363 $\mathrm{mm\,year^{-1}}$, while $E_s$ increased by 17% from 156 to 183 $\mathrm{mm\,year^{-1}}$. In this region, the total evaporation reduced only by 1% from 1174 to 1162 $\mathrm{mm\,year^{-1}}$ in IALC compared to CTR, because the reductions in $E_t$ and $E_i$ were partially compensated by increased $E_s$. The reduced $E$ in IALC is closer to the the DOLCEv3 $E$, which is in this region 1160 $\mathrm{mm\,year^{-1}}$. We also found the evaporation compensation effect in Lapland, where $A_H$ increased from 0.24 to 0.30, and Central Asia, where $A_L$ increased from 0.66 to 0.71 (Fig. 3). In Lapland and in Central Asia $E$ increased with 2% and 0.1%, respectively, moving closer to the DOLCEv3 $E$ (Fig. 5b; Table S2). In contrast with the small changes in $E$, the individual $E$-fluxes changed considerably in these two cases (Fig. 5d,f,h).

The changes in $E_t$ and $E_s$ also induced changes in soil moisture, because $E_s$ extracts water exclusively from the near-surface soil layer ($\mathrm{SM_s}$), while $E_t$ originates mostly from deeper soil layers ($\mathrm{SM_{sb}}$). However, we observed only marginal differences between mean $\mathrm{SM_s}$ and $\mathrm{SM_{sb}}$ in IALC compared to CTR (Fig. 6), except for the Southern Amazon $\mathrm{SM_s}$. Here, the increased $E_s$ in the Southern Amazon reduced the $\mathrm{SM_s}$ by 2%, as more near-surface soil moisture was extracted (Fig. 6b). Individual evaporation fluxes influence the timing of total evaporation and soil moisture differently. However, the overall minor magnitude of changes in total E and $\mathrm{SM_s}$ in IALC compared to CTR led to marginal changes in RMSE and pearson correlation coefficients with respect to the reference data in the three highlighted cases (Table S3, Fig. S1-S3).

## 3.2 Leaf Area Index inter-annual variability effects

The inclusion of inter-annual LAI variability in IAK5 (Fig. 4) generally led to an increased anomaly standard deviation (i.e. variability) of $E$ (Fig. 7a,b). This effect is mostly dominated by $E_t$ (Fig. 7d), which, in the model, is linearly related to LAI (Eq. (1)). Figure 7d and h show that the variability in $E_t$ and $E_i$ were mostly increased in semiarid regions such as the Great Plains region of the US, Central Asia and Southern Africa, with a stronger effect for $E_t$ than for $E_i$. In contrast, the $E_s$ variability was reduced with the enhanced LAI variability in these semiarid regions, but was increased in more temperate regions such as in Europe, the Eastern US and the La Plata Basin in South America (Fig. 7e,f). While the $E_t$ anomaly variability considerably increased in IAK5 compared to IALC in semiarid regions, the anomaly variability in subsurface soil moisture ($\mathrm{SM_{sb}}$), that acts as the main source of $E_t$, reduced in these regions (Fig. 8c,d). On the other hand, the anomaly variability of $\mathrm{SM_s}$ increased (Fig. 8a,b), while the $E_s$ variability reduced.

Figure 9 shows that pearson correlation coefficient ($r$) of anomaly $E$ with respect to DOLCEv3 increased in IAK5 compared to IALC in 85% of the land area in which the $r$ was significantly different in IAK5 compared to IALC. Consistently, the $r$ of



anomaly $SM_s$ with respect to ESA-CCI SM also improved in 85% of the significantly changing land area. For both E and $SM_s$
$r$ increased mostly in semiarid regions with predominantly low vegetation (Fig. 3).

### 3.3 Vegetation specific effective vegetation cover parameterization effects

The observed relationship of LAI and FCover in Fig. 10 is broadly consistent with the shape of the exponential functions with
the vegetation specific $k$, with RMSEs between 0.018 and 0.053 for the individual vegetation types. All optimized LAI-FCover
relations are characterized by $k$-values that are with 0.351-0.458 consistently lower than the original $k$=0.5, which has been
used as constant default value in most HTESSEL applications so far (Alessandri et al., 2017; Boussetta et al., 2021). We found
that the $k$-values for low vegetation types (0.438-0.458) are higher than for high vegetation types (0.351-0.396), except for
tundra regions (0.375).

The vegetation specific $C_{\mathrm{eff}}$ parameterization (IAKV) generally reduced the $k$-values compared to the $k$=0.5 setup (IAK5),
and as a consequence the associated vegetation densities $c_{\mathrm{v,L}}$ and $c_{\mathrm{v,H}}$ also decreased (Eqs. (8) and (9)). On average, the global
mean $C_{\mathrm{eff}}$ reduced from 0.21 in IAK5 to 0.19 in IAKV (Fig. 11). The reduced $C_{\mathrm{eff}}$ considerably reduced the RMSE with
respect to the FCover data in IAKV compared to IAK5 (Fig. 12), as expected from the parameterization optimization presented
in Fig. 10. The RMSE reduced the most over the boreal and tropical forests, with an average RMSE reduction from 0.12 to 0.06
for evergreen needleleaf trees, and from 0.05 to 0.03 for evergreen broadleaf trees. On the other hand, the differences in regions
with predominantly low vegetation were smaller because the fitted $k$-value was closer to the original $k$=0.5, with an average
RMSE reduction from 0.06 in IAK5 to 0.05 in IAKV for crops, and from 0.04 to 0.03 for short grass. For low vegetation, the
effects were not consistent throughout the seasons, with RMSE increasing at high latitudes in JJA (Fig. 12d). Here the $C_{\mathrm{eff}}$ in
IAK5 was smaller than the CGLS FCover and is further reduced in IAKV, increasing the RMSE. This was likely caused by a
poor fit for short grass at LAI >2 (Fig. 10b) and tundra at LAI >1 (Fig. 10g).

The reduced model $C_{\mathrm{eff}}$ in IAKV compared to IAK5 led to a shift in individual evaporation fluxes. On average, $E_s$ increased
and $E_t$ and $E_i$ reduced, while the total $E$ was not much affected (Fig. S6). These shifts led to changes in the temporal
distribution of the evaporation. Figure 13 shows quite mixed results of the vegetation specific $C_{\mathrm{eff}}$ parameterization on the
model $E$ RMSE with respect to DOLCEv3. The RMSE consistently reduced during summer months in temperate regions such
as in Europe, Eastern US and Eastern China (JJA), and in South-Eastern Brazil and Southern Africa (DJF). On the other hand,
the results for tropical and boreal regions were less consistent throughout the seasons (Fig. 13). The effects of the vegetation
specific $C_{\mathrm{eff}}$ on $SM_s$ RMSE with respect to ESA-CCI SM show consistent RMSE reductions in the JJA period for Canada and
South-Eastern Brazil, and in the DJF period for the Sahel (Fig. 14). Consistent with the $C_{\mathrm{eff}}$ RMSE increase in boreal regions
in JJA (Fig. 12), the Pearson correlation coefficient for monthly anomaly $E$ with respect to DOLCEv3 $E$ significantly reduced
in these regions in IAKV compared to IAK5, while other regions were not much affected (Fig. S14). On the other hand, the
correlation of monthly anomaly $SM_s$ with respect to ESA-CCI SM did not considerably change (Fig. S14).





### 3.4 Combined effects of land cover, Leaf Area Index, and effective vegetation cover


The results presented in Sect.3.2 demonstrate that the inter-annually varying LAI in experiment IAK5 considerably improved the correlation of $E$ and $\mathrm{SM_s}$ with respect to reference data. On the other hand, the annually varying LC and vegetation specific $C_\mathrm{eff}$ affected correlations merely to a minor degree (Sects.3.1 and 3.3). Here, we further elaborate on the effects of combining the enhanced variability in LC, LAI and $C_\mathrm{eff}$ on correlation of $E$ and $\mathrm{SM_s}$.

Figure 15 shows that the $E$ correlation improved in 68 % (JJA) and 54 % (DJF) of the land area in which the $r$ significantly changed in IAKV compared to CTR. Significant reduction of $r$ is found over boreal regions, which is related to the effects of the effective vegetation cover parameterization, as discussed in Sect. 3.3 and Fig. S14. Figures 15b and d show that the $\mathrm{SM_s}$ correlation consistently and significantly improved in 83 % (JJA) and 76 % (DJF) of the land area in which the $r$ significantly changed in IAKV compared to CTR. The $E$ and $\mathrm{SM_s}$ correlations got consistently stronger during dry periods in regions

with a semiarid climate and predominantly low vegetation (Figs. 3 and 15). For example, in North-Eastern Brazil during the dry JJA season, the correlation coefficient for E increased from $r=0.79$ in CTR to 0.84 in IAKV with respect to DOLCEv3 and for $\mathrm{SM_s}$ from $r=0.57$ to 0.67 with respect to ESA-CCI SM. Similarly, in Western India during the dry DJF season, the correlation coefficient for $E$ increased from $r=0.78$ to 0.85, and for $\mathrm{SM_s}$ from $r=0.45$ to 0.73. To further explore the effects in these semiarid regions, we zoom in to North-Eastern Brazil for the 2010-2013 period (Fig. 16). This period is characterized

by positive LAI and $C_\mathrm{eff}$ anomalies in JJA 2011 and a negative LAI and $C_\mathrm{eff}$ anomalies in JJA 2012 (Fig. 16a,b). The negative LAI and $C_\mathrm{eff}$ anomalies in 2012 characterize a dry period in which the negative $E$ anomaly was magnified in IAKV compared to CTR (Fig. 16c). During this dry period, $E_\mathrm{t}$ reduced, while $E_\mathrm{s}$ increased. This is consistent with the soil moisture response presented in Fig. 16d, as the $\mathrm{SM_s}$ reduced (due to more $E_\mathrm{s}$) and the $\mathrm{SM_{sb}}$ increased (due to less $E_\mathrm{t}$) during the 2012 dry period. Opposite effects were found for the 2011 period with positive LAI and $C_\mathrm{eff}$ anomalies. So in this specific case, the variability

in $E_\mathrm{t}$ and $\mathrm{SM_s}$ anomalies was enhanced in IAKV compared to CTR, while the variability in $E_\mathrm{s}$ and $\mathrm{SM_{sb}}$ anomalies was dampened. This is consistent with the results presented in Figs. 7 and 8, in which the effect of the inter-annually varying LAI on the variability of $E$ and SM are presented.

The opposing effects of the enhanced LAI variability on anomaly $E_\mathrm{t}$ and $\mathrm{SM_{sb}}$ can be explained by a negative feedback between vegetation and soil moisture schematized on the right side in Fig. 17a. During dry periods, the soil moisture reduces;

this lower soil water availability can result in vegetation water stress; consequently leading to lower vegetation activity in terms of transpiration and primary production, which is reflected, for example, in the typical dry season browning of grass species in low-vegetation regions and in the model represented by negative LAI and $C_\mathrm{eff}$ anomalies (Fig. 16a,b). As transpiration is reduced (Fig. 16c), the negative sub-surface soil moisture anomaly is similarly reduced, because less water is extracted (Fig. 16d). On the other hand, the enhanced vegetation variability activated a positive feedback between anomaly vegetation activity

and anomaly $\mathrm{SM_s}$, as illustrated on the left side of Fig. 17a. Reduced vegetation activity is reflected in the model by a reduced $C_\mathrm{eff}$ and an increased bare soil fraction (Eq. (7)), which leads to an increased $E_\mathrm{s}$, (Fig. 16b), and, as a consequence, less $\mathrm{SM_s}$ during a dry period as long as soil moisture is available (Fig. 16d).





Figures 17b and c show that the positive feedback between $E_\mathrm{s}$ and $\mathrm{SM_s}$, as introduced by the improved vegetation variability, is the strongest over semiarid regions with low vegetation, while the negative feedback between $E_\mathrm{t}$ and $\mathrm{SM_{sb}}$ is more

pronounced for temperate regions with deciduous vegetation and crops, where the inter-annual LAI variability is larger (Fig. 4).

## 4   Discussion

### 4.1   Synthesis of results

The results presented in Sect. 3.4 indicate overall improvements of correlation coefficients of $E$ and $\mathrm{SM_s}$ with all three aspects

of vegetation variability implemented. We attribute these effects primarily to the implementation of inter-annually varying LAI, as the effects of the LC variability and vegetation specific $C_\mathrm{eff}$ on $E$ and $\mathrm{SM_s}$ were smaller (Sects. 3.1 and 3.3). The pronounced improvements in $\mathrm{SM_s}$ and $E$ correlation in semiarid regions (Fig. 15) are directly related to the feedback mechanisms typical of water-limited regions that were activated by the vegetation variability. Regions where the positive feedback is strong (Fig. 17b) coincide with the regions that exhibit a strengthening of the correlations. In the model setup with seasonally varying LAI

only (experiments CTR and IALC), the feedbacks in Fig. 17 are not represented because the interaction between SM and LAI is activated by the inter-annually varying LAI. In particular, the interactions between LAI, $C_\mathrm{eff}$ and bare soil cover are only captured if model $C_\mathrm{eff}$ is exponentially related to LAI (Sect. 2.2.3). This finding complements the arguments from previous studies for using the exponential LAI-$C_\mathrm{eff}$ relation instead of the lookup table $C_\mathrm{eff}$ in HTESSEL (Alessandri et al., 2017; Johannsen et al., 2019; Nogueira et al., 2020, 2021).

The vegetation specific effective vegetation cover parameterization presented in Sect. 3.3 generally resulted in an improved match of model $C_\mathrm{eff}$ and CGLS FCover (Fig. 12), which was expected because the FCover data was used for the estimation of the exponential coefficient $k$ based on least squares minimization (Sect. 2.3.3). CGLS FCover explicitly represents the fraction of green vegetation cover, and, therefore, matches the model actively transpiring vegetation fraction $C_\mathrm{eff}$. However, also the non-green vegetated area cover affects the atmosphere by e.g. modifying albedo and roughness lengths, which is not

considered in the model, as non-green vegetation is represented as bare soil. This is a limitation for the present implementation of the vegetation specific effective vegetation cover parameterization. The results presented in Fig. 13 showed both increased and reduced RMSE for $E$ with respect to the reference data in IAKV compared to IAK5. Consistent reductions of $E$ RMSE for in Europe and Eastern US in the JJA period were found. These regions coincide with regions with a high density of FLUXNET tower observations used for generation of the DOLCEv3 $E$ (Hobeichi et al., 2021). The lack of tower observations in the

Tropics, the Sahel, South Eastern Asia and at high latitudes may potentially explain the mixed RMSE results in these regions presented in Fig. 13. For high latitudes (e.g. Northern Canada and Eastern Siberia) the RMSE for both $E$ as $C_\mathrm{eff}$ increased and the pearson correlation reduced (Fig. S14) in IAKV compared to IAK5 for the JJA period. This might be at least in part related to the poor fit of the parameterization for high LAI values for short grass and tundra, as explained in Sect. 3.3 (Fig. 10).

The inter-annually varying land cover locally affected the model $E$ and SM as expected, with reduced (increased) $E$ driven

by corresponding reductions (increases) in high vegetation cover fraction (Fig. 5 and 6). However, the effects on $E$ and SM





are likely underestimated due to the HTESSEL land cover structure in which the dominant vegetation type and cover fraction are used, and vegetation mixing within high or low vegetation types is not represented (Fig. 1). With this, only major changes in the ESA-CCI vegetation types and fractions are captured by the model. In IALC we evaluated the effects of inter-annually varying LC individually, but for internal consistency LAI and LC inter-annual variations should ideally be used together as they

are interdependent. The local effects of the inter-annually varying land cover on the total $E$ were considerably smaller than on the individual $E$ fluxes (Fig. 5). The reduced (increased) $E_t$ and $E_i$, were compensated for by increased (reduced) $E_s$. This compensation is related to the $C_{eff}$ parameterization (Eq. (6)), and also to the offline setup, which does not allow for couplings with the atmosphere. Reduced $A_H$ in the Amazon (Fig. 3), led to a reduced $C_{eff}$ and an increased bare soil fraction (Eq. (4)-(7)), and, therefore, reduced $E_t$ and $E_i$, and increased $E_s$, in order to fulfill the atmospheric evaporation demand defined by

the prescribed atmospheric forcing. Similarly, the on average reduced $C_{eff}$ with the vegetation specific $C_{eff}$ parameterization (Fig. 11) introduced in experiment IAKV, led to a shift in annual mean individual $E$-fluxes, with increased $E_s$ and reduced $E_t$ and $E_i$ (Fig. S6).

It is important to note that the partitioning of evaporation into the three individual components $E_t$, $E_s$ and $E_i$ in the model remains problematic to compare with observations. There is wide-spread consensus that, globally averaged, transpiration is the

largest land evaporation flux component, followed by soil evaporation and interception evaporation (Miralles et al., 2011; Wei et al., 2017; Nelson et al., 2020). However, estimates of the average global $E_t$ contribution to total terrestrial evaporation are subject to major uncertainties, with the global $E_t$ contribution quantified in the range of 35-80 % (Schlesinger and Jasechko, 2014; Coenders-Gerrits et al., 2014). The global mean modelled partitioning of evaporation in our study is on the low end of these estimates with 39 % $E_t$, 38 % $E_s$ and 20 % $E_i$ in CTR and 38 % $E_t$, 41 % $E_s$ and 20 % $E_i$, in IAKV (the values do not

add to 100 % due to open water evaporation). Despite the consistent improvements in anomaly correlation coefficients of $E$ and $SM_s$ found in IAKV compared to CTR (Fig. 15), the apparently low contribution of $E_t$ to total $E$ needs further evaluation, which was out of scope in this study.

## 4.2 Methodological limitations

Our model experiments were performed in an offline mode with fixed atmospheric forcing, which allowed us to analyse

individual hydrological processes in detail. However, the fixed atmospheric model input considerably constrains changes of model surface fluxes. Moreover, the ERA5 forcing used here is based on a LSM that does not represent land cover and vegetation variability, which is partially corrected for by data assimilation of observations (Hersbach et al., 2020; Nogueira et al., 2021). The potential mismatch between our LSM and the ERA5 atmospheric forcing may also have influenced the observed model effects. Another possible limitation is the absence of re-calibration of model parameters, such as roughness

lengths and minimum stomatal resistances. Fixed model parameters were originally calibrated using the lookup table $C_{eff}$ parameterization, MODIS LAI and GLCC LC, and have not been adjusted for the three new model scenarios tested here (IALC, IAK5 and IAKV). This was also emphasized by Johannsen et al. (2019), Nogueira et al. (2020, 2021) and Boussetta et al. (2021) who concluded that model vegetation changes should be implemented in an integral context and re-calibration of model parameters is needed.



This study emphasizes the importance of realistic representation of vegetation variability for modelling land surface-atmosphere interactions. However, for further applications it is needed to explore how the vegetation variability influences atmospheric variables in a coupled model setup. The availability of reliable observations is therefore fundamental to properly understand and model the processes of relevance for land surface and the interaction with atmosphere. Here, the evaluation of model performance was limited to total evaporation and near-surface soil moisture. The evaluated performances of model $E$ and $\mathrm{SM_s}$ need

to be interpreted in a careful way, bearing in mind the uncertainties. For total evaporation we used the DOLCEv3 evaporation data that merges FLUXNET tower observations with evaporation from FLUXCOM-RSMETEO, GLEAM v3.5a and v3.5b and ERA5-land, which all include very specific model assumptions on vegetation representations. Although these data are considered suitable for time series and trend analyses, the associated uncertainty estimates are large (Hobeichi et al., 2021) (Fig. 16). Figure 16c shows that the DOLCEv3 inter-annual variability is systematically smaller than the modeled variability. This

limited inter-annual variability in DOLCEv3 could be at least in part related to the combination of several products, because the averaging based on FLUXNET towers in-avoidably dampens the anomalies, reducing the inter-annual variability. Evaluation of the modeled near-surface soil moisture was limited by missing data due to dense forests or snow cover, and the lack of detailed information of the representative soil depth. However, we still find the ESA-CCI SM the most trustworthy reference data because it is a direct product of remote sensing observations, without blending land surface models as done for DOLCEv3.

**5   Conclusions**

This study aimed to address the limitations of state-of-the-art land surface models in representing spatial and temporal vegetation dynamics. We evaluated the effects of improving the representation of land cover and vegetation variability based on satellite observational products in the HTESSEL land surface model. Specifically, we directly integrated satellite based inter-annually varying land cover, and seasonally and inter-annually varying LAI. In addition, we formulated and integrated

an effective vegetation cover parameterization that can distinguish between different vegetation types. The effects of these three implementations were analysed for soil moisture and evaporation in offline experiments forced with ERA5 atmospheric forcing.

The inter-annually varying land cover locally altered the model evaporation and soil moisture. In regions with major land cover changes, such as the Amazon, the model evaporation fluxes and soil moisture responded consistently, capturing the

effects of increased or decreased high or low vegetation cover. The inter-annually varying LAI led to significant improvements of the correlation coefficients computed with the available observations of near-surface soil moisture and evaporation. This was specifically true in semiarid regions with predominantly low vegetation, during the dry season. The inter-annually varying LAI and effective vegetation cover allow for an adequate representation of soil moisture-evaporation feedback by activating the couplings with vegetation during vegetation-water-stressed periods (Fig. 17). From these results, we conclude that it is

essential to realistically represent inter-annual variability of LAI, and to include the exponential relation between LAI and effective vegetation cover to correctly capture land-atmospheric feedbacks during droughts in HTESSEL. The developments of the effective vegetation cover parameterization considerably improved the spatial and temporal variability of the model

effective vegetation cover, and regionally reduced the model errors of evaporation and near-surface soil moisture. Overall, our results emphasize the need for representing spatial and temporal vegetation variability in LSMs used for climate reanalyses
and near-term climate predictions. In climate predictions, we obviously cannot rely on satellite observations, and, therefore, the development and validation of dynamical or statistical models able to reliably predict vegetation evolution is an important challenge for the land surface modelling community.

*Code and data availability.* ESA-CCI land cover data was taken from https://cds.climate.copernicus.eu/cdsapp#!/dataset/satellite-land-cover? tab=overview (Copernicus Climate Change Service, 2019) (last access: June 2021). CGLS LAI and FCover data were downloaded from
https://land.copernicus.eu/global/products/ (last access: January 2022) (Copernicus Global Land Service, 2022). AVHRR LAI was accessed from https://www.ncei.noaa.gov/access/metadata/landing-page/bin/iso?id=gov.noaa.ncdc:C01559 (last access: June 2021). The preparation of the ESA-CCI land cover, CGLS LAI and AVHRR LAI for the use in HTESSEL is explained in Boussetta and Balsamo (2021). DOLCEv3 was accessed through https://doi.org/10.25914/606e9120c5ebe (last access: May 2022) (Hobeichi et al., 2021) and ESA-CCI SM through https://esa-soilmoisture-cci.org/data (last access: May 2022). The offline HTESSEL model was provided by EC-EARTH, together with the
ERA5 forcing data as well as vegetation and soil data. The python scripts used for the LAI-FCover non linear least squared optimization, and for the analyses are available on github https://github.com/fvanoorschot/lai_fcover_fitting and https://github.com/fvanoorschot/htessel_ analyses. Upon acceptance, these scripts will be moved to Zenodo with a doi assigned to them, and the underlying data will be shared in an open research database.

*Author contributions.* The study was conceived by AA and amended with input from all co-authors. FO carried out the study and prepared
the original draft with supervision of AA, RE and MH. SB and GB prepared part of the data used for modeling. All authors contributed to review and editing of the final draft.

*Competing interests.* The authors declare that they have no conflict of interest

*Acknowledgements.* Acknowledgement is given for the use of ECMWF's computing and archive facilities for this research, which were
provided by CNR-ISAC and by ECMWF in the framework of the special project SPITALES. This work was supported by the European Union's Horizon 2020 research and innovation program under grant agreement no. 101004156 (CONFESS project). Ruud J. van der Ent acknowledges funding from the Netherlands Organization for Scientific Research (NWO) under project number 016.Veni.181.015.



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



**Table 1.** Details of model experiments

| Experiment | Land cover | Leaf Area Index | Effective vegetation cover |
|---|---|---|---|
| CTR | ESA-CCI fixed | CGLS climatology | $k$=0.5 |
| IALC | ESA-CCI inter-annual | CGLS climatology | $k$=0.5 |
| IAK5 | ESA-CCI inter-annual | CGLS inter-annual | $k$=0.5 |
| IAKV | ESA-CCI inter-annual | CGLS inter-annual | $k$ vegetation specific |





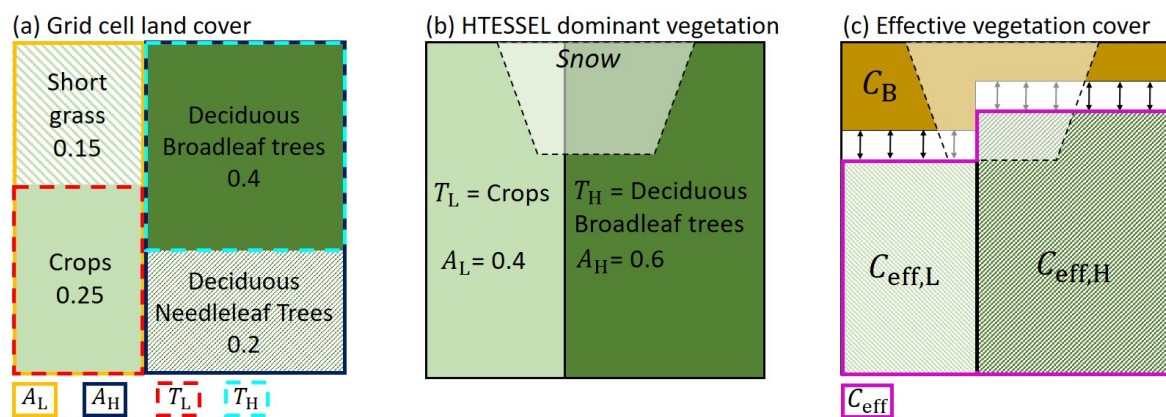

**Figure 1.** Vegetation representation in a grid cell with example vegetation types and cover fractions. (a) Grid cell vegetation type and cover fraction based on land cover dataset. (b) HTESSEL dominant low and high vegetation type ($T_\mathrm{L}$ and $T_\mathrm{H}$) and cover fraction ($A_\mathrm{L}$ and $A_\mathrm{H}$). (c) HTESSEL effective vegetation cover with $C_\mathrm{eff,L}$ and $C_\mathrm{eff,H}$ the effective low and high vegetation cover fraction, $C_\mathrm{B}$ the bare soil fraction, and $C_\mathrm{eff} = C_\mathrm{eff,L} + C_\mathrm{eff,H}$, with the arrows indicating the temporal variability of $C_\mathrm{eff}$ as discussed in Sect. 2.2.3.

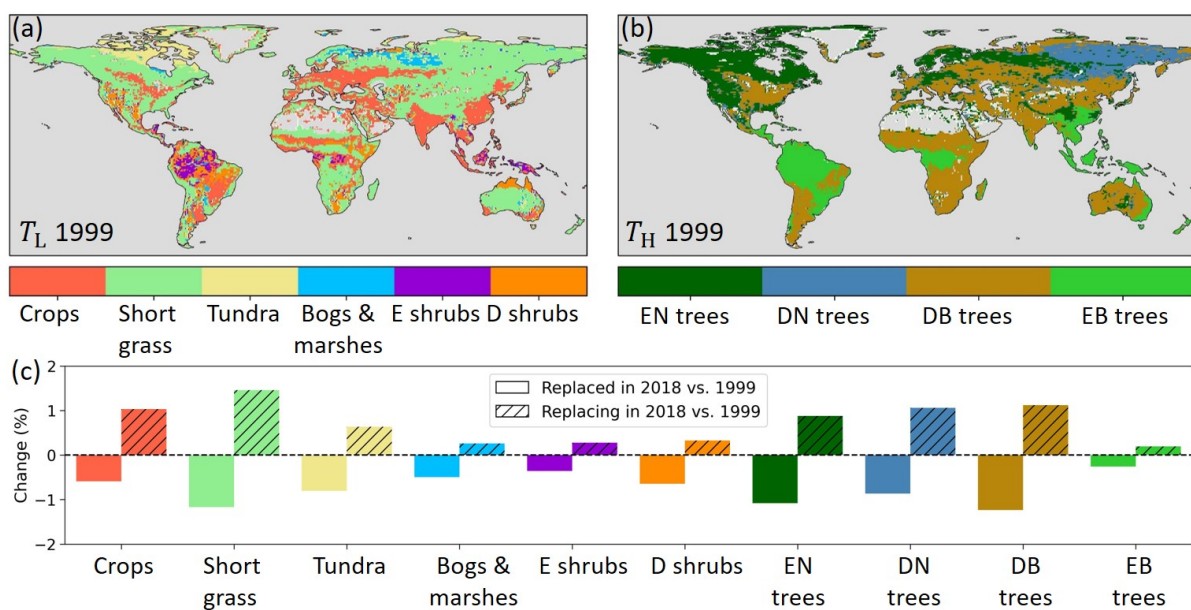

**Figure 2.** (a) Model low ($T_L$) and (b) high ($T_H$) dominant vegetation types in 1999 based on ESA-CCI land cover. (c) Changes in vegetation type in percent of the total land points with plain colors indicating that the vegetation type has been replaced in 2018 compared to 1999 and hatched colors that the vegetation replaces another type in 2018 compared to 1999. E stands for evergreen, D for deciduous, N for needleleaf, and B for broadleaf

.





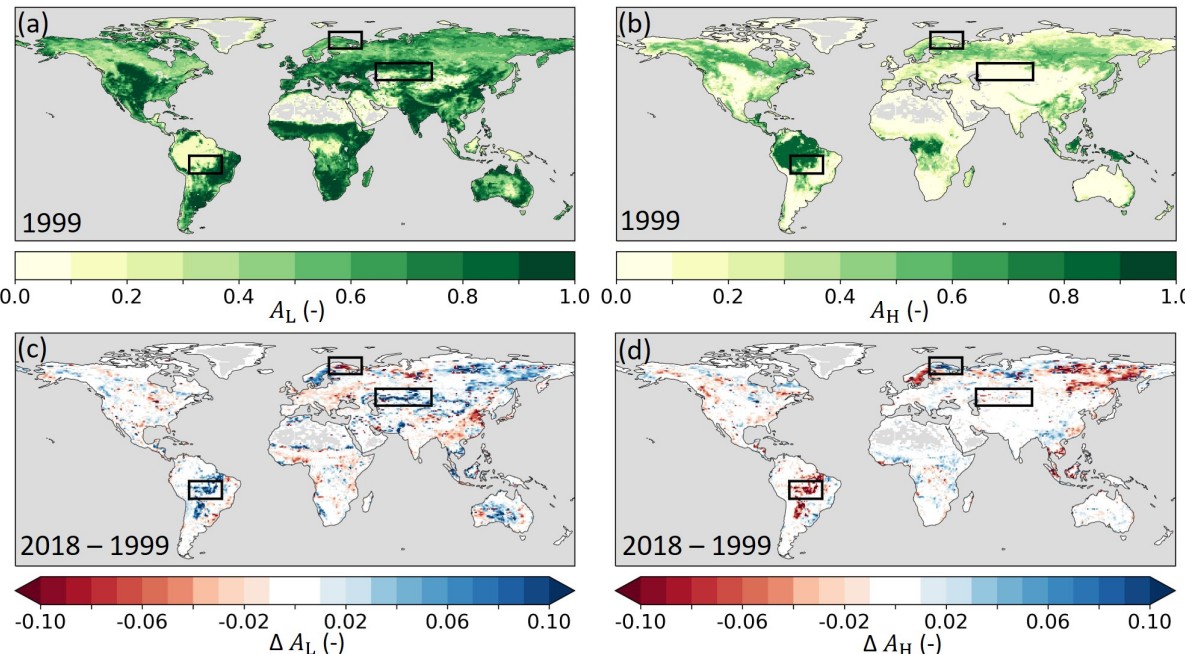

**Figure 3.** (a) Model low ($A_\mathrm{L}$) and (b) high ($A_\mathrm{H}$) vegetation cover fraction in 1999 and absolute difference in (c) $A_\mathrm{L}$ and (d) $A_\mathrm{H}$ between 2018 and 1999 (2018−1999) based on ESA-CCI land cover. Blue (red) indicates an increased (reduced) cover in 2018. The black boxes highlight the three regions Southern Amazon, Lapland and Central Asia that are further analyzed in Sect. 3.1.



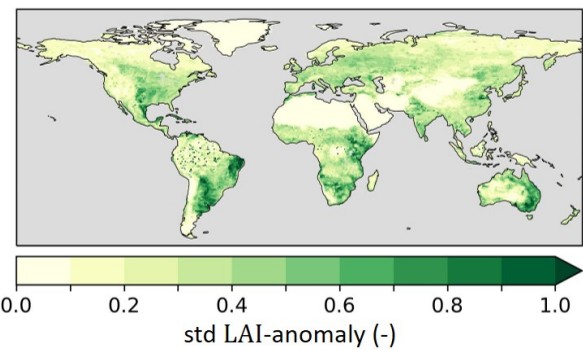

**Figure 4.** Standard deviation (std) of monthly inter-annual anomaly CGLS LAI for 1999-2018 as implemented in experiment IAK5 (Table 1).





**Figure 5.** Annual mean evaporation fluxes (2014-2018) in experiment CTR with (a) total evaporation ($E$), (c) transpiration ($E_t$), (e) soil evaporation ($E_s$) and (g) interception evaporation ($E_i$), and the relative difference ($\Delta_{rel}$) between annual mean evaporation fluxes in experiments IALC and CTR ((IALC−CTR)/CTR) for (b) $E$, (d) $E_t$, (f) $E_s$ and (h) $E_i$. Blue (red) indicates an increased (reduced) flux. Grey land areas indicate regions with annual mean E-fluxes < 0.1 mm year$^{-1}$. The boxes highlight the three regions Southern Amazon, Lapland and Central Asia with major land cover changes (Fig. 3). See Table 1 for details of the experiments.



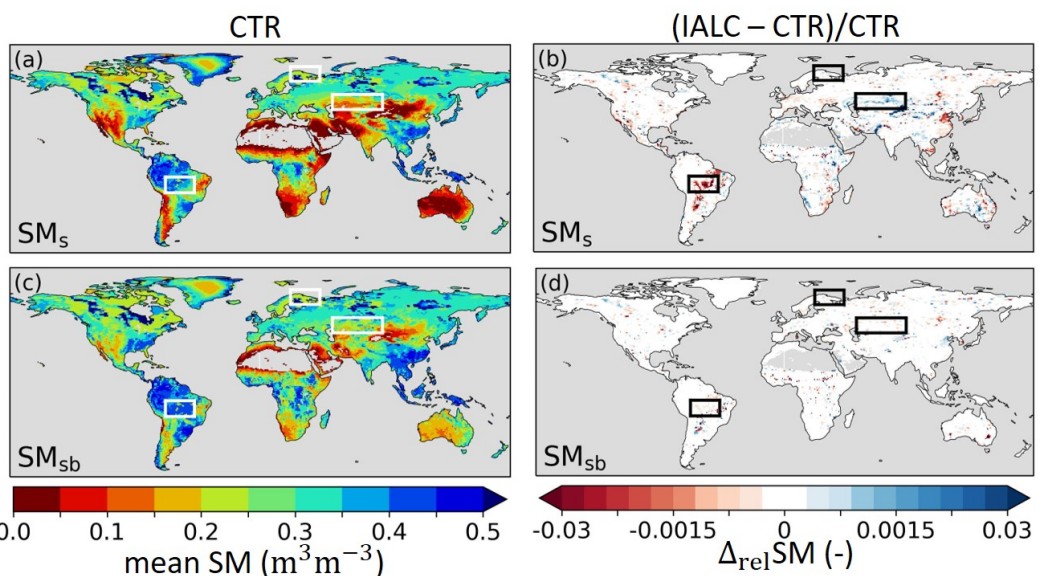

**Figure 6.** Annual mean soil moisture (2014-2018) (SM) in experiment CTR with (a) near-surface soil moisture ($SM_s$) and (c) subsurface soil moisture ($SM_{sb}$), and the relative difference ($\Delta_{rel}$) between annual mean SM in experiments IALC and CTR ((IALC−CTR)/CTR) for (b) $SM_s$ and (d) $SM_{sb}$. Blue (red) indicates an increased (reduced) soil moisture. Grey land areas indicate regions with annual mean SM < $0.01\,\mathrm{m^3\,m^{-3}}$. The boxes highlight the three regions Southern Amazon, Lapland and Central Asia with major land cover changes (Fig. 3). See Table 1 for details of the experiments.





**Figure 7.** Standard deviation (std) of anomaly evaporation fluxes in experiment IALC with (a) total evaporation ($E$), (c) transpiration ($E_t$), (e) soil evaporation ($E_s$) and (g) interception evaporation ($E_i$), and the relative difference ($\Delta_{rel}$) between the anomaly E std in experiments IAK5 and IALC ((IAK5−IALC)/IALC) for (b) $E$, (d) $E_t$, (f) $E_s$ and (h) $E_i$. Blue (red) indicates an increased (reduced) std. See Table 1 for details of the experiments.

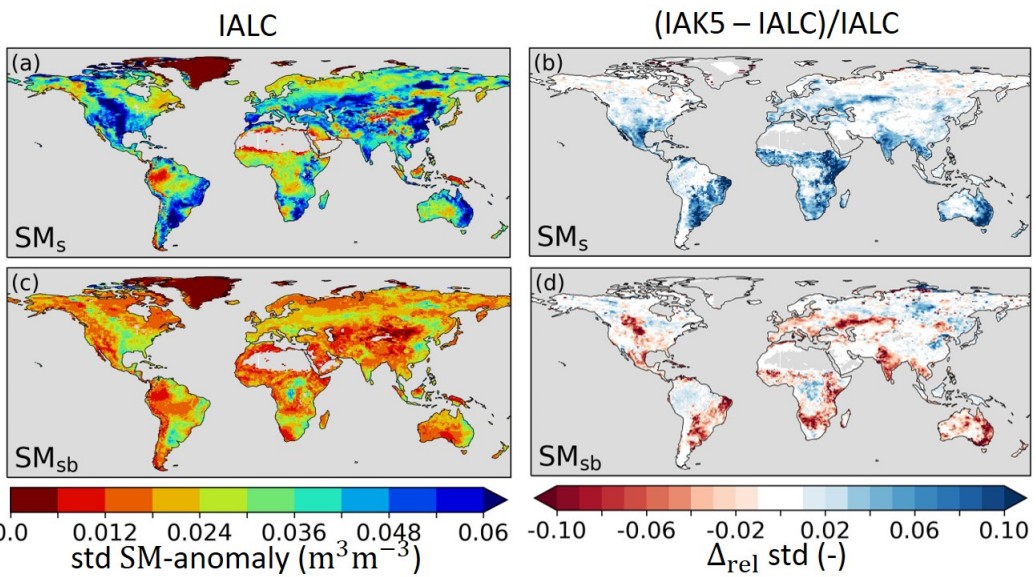

**Figure 8.** Standard deviation (std) of anomaly soil moisture (SM) in experiment IALC with (a) near-surface soil moisture ($SM_s$), and (c) subsurface soil moisture ($SM_{sb}$), and the relative difference ($\Delta_{rel}$) between the anomaly SM std in experiments IAK5 and IALC ((IAK5−IALC)/IALC) for (b) $SM_s$ and (d) $SM_{sb}$. Blue (red) indicates an increased (reduced) variability. See Table 1 for details of the experiments.

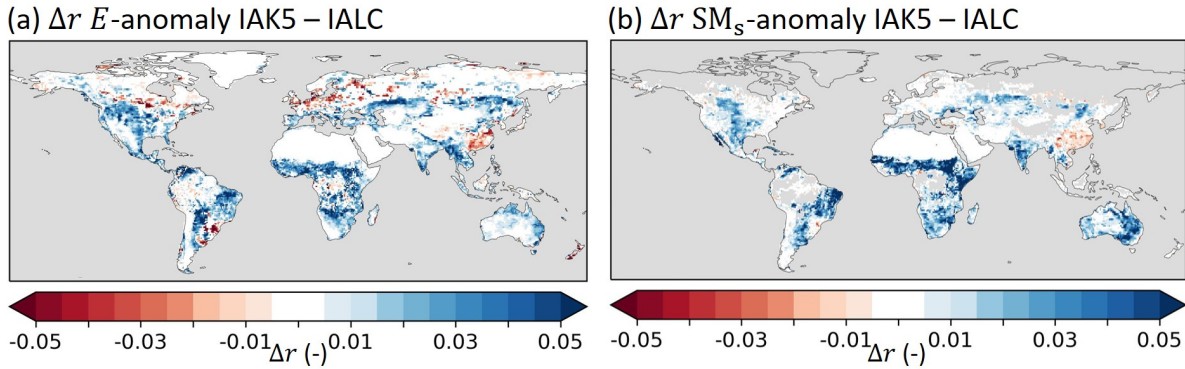

**Figure 9.** Pearson correlation difference ($\Delta r$) between experiments IALC and IAK5 (IAK5−IALC) for (a) monthly anomaly total evaporation ($E$) with respect to DOLCEv3 evaporation and (b) monthly anomaly near-surface soil moisture ($\mathrm{SM_s}$) with respect to ESA-CCI SM. Blue (red) indicates an increased (reduced) correlation in IAK5 compared to IALC, white colors indicate small and/or insignificant $\Delta r$, and grey indicates no data points. See Table 1 for details of the experiments. Similar figures for seasonal anomalies are presented in Fig. S4-5.



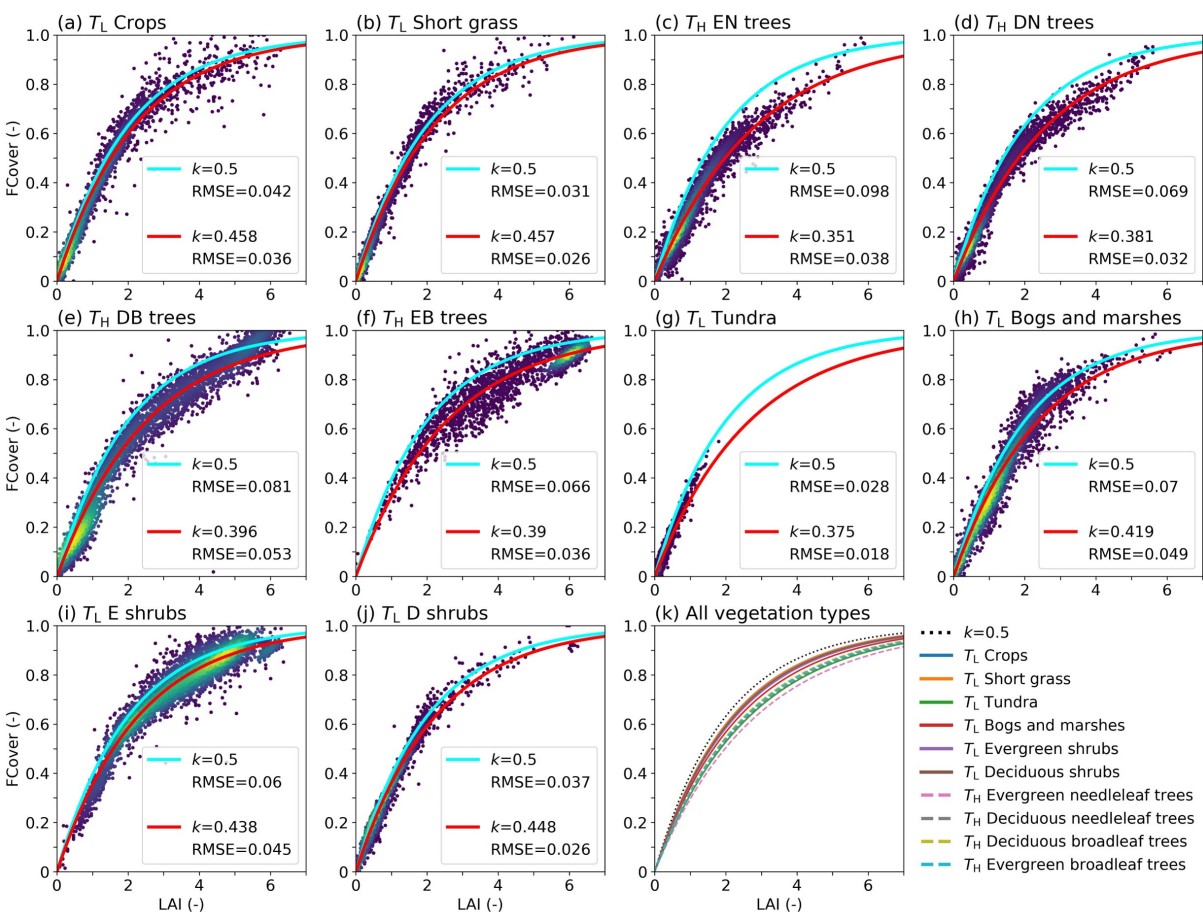

**Figure 10.** (a – j) LAI vs FCover for a subsample (5000) of the selected points used for the least squares optimization for all vegetation types with in red the optimized LAI-FCover relation (Eq. 10) and in lightblue the $k$=0.5 relation, with RMSE values of the data points with respect to the curve. The colors indicate the point density with purple a low density and yellow a high density. (k) The optimized LAI-FCover relation for all vegetation types. E stands for evergreen, D for deciduous, N for needleleaf, and B for broadleaf.



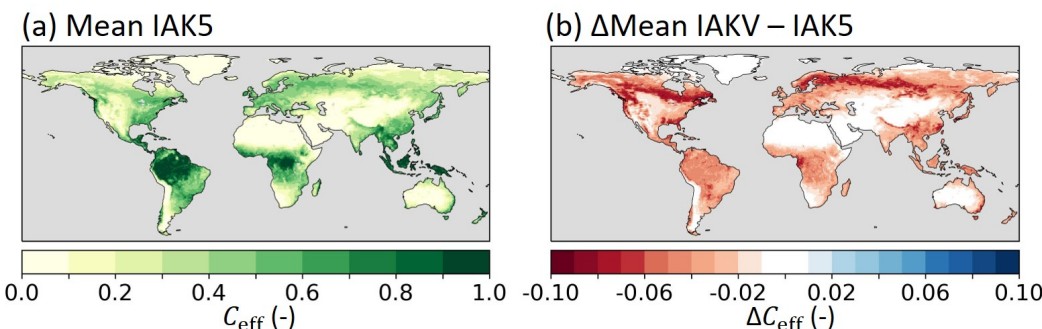

**Figure 11.** (a) Mean monthly model effective vegetation cover ($C_{\text{eff}}$) in experiment IAK5 and (b) the absolute difference between IAKV and IAK5 (IAKV−IAK5) mean monthly $C_{\text{eff}}$, red (blue) indicates a reduced (increased) $C_{\text{eff}}$ in IAKV compared to IAK5. Details of model experiments in Table 1.

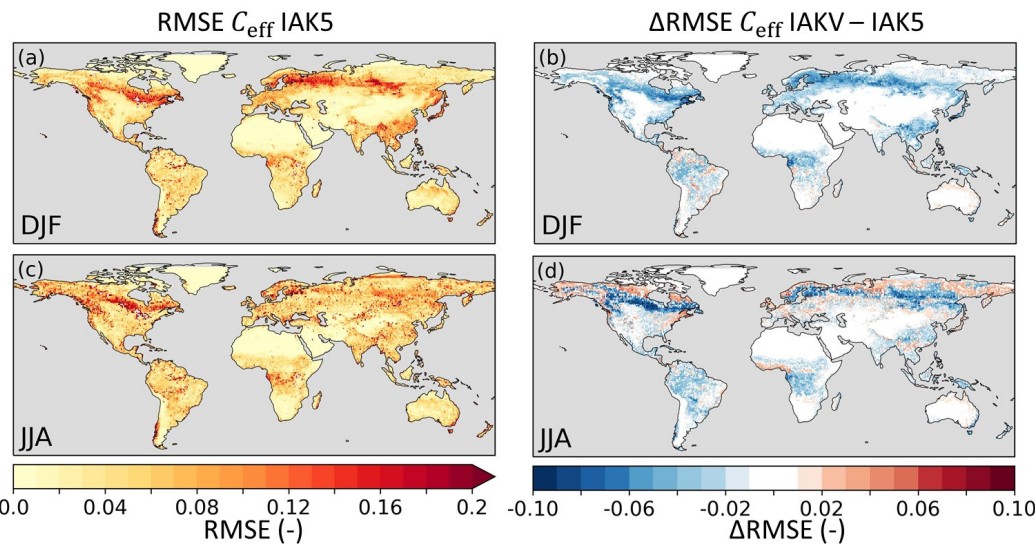

**Figure 12.** Root mean squared error (RMSE) of model seasonal $C_{\mathrm{eff}}$ in experiment IAK5 with respect to CGLS FCover for DJF (a) and JJA (c), with red indicating a larger RMSE. The difference between RMSE in IAK5 and IAKV (IAKV−IAK5) for DJF (b) and JJA (d) with blue (red) indicating a reduced (increased) RMSE, and white colors indicate small and/or insignificant ΔRMSE. See Table 1 for details of the experiments. Similar figures for monthly values and all the seasons are presented in Fig. S8-9.



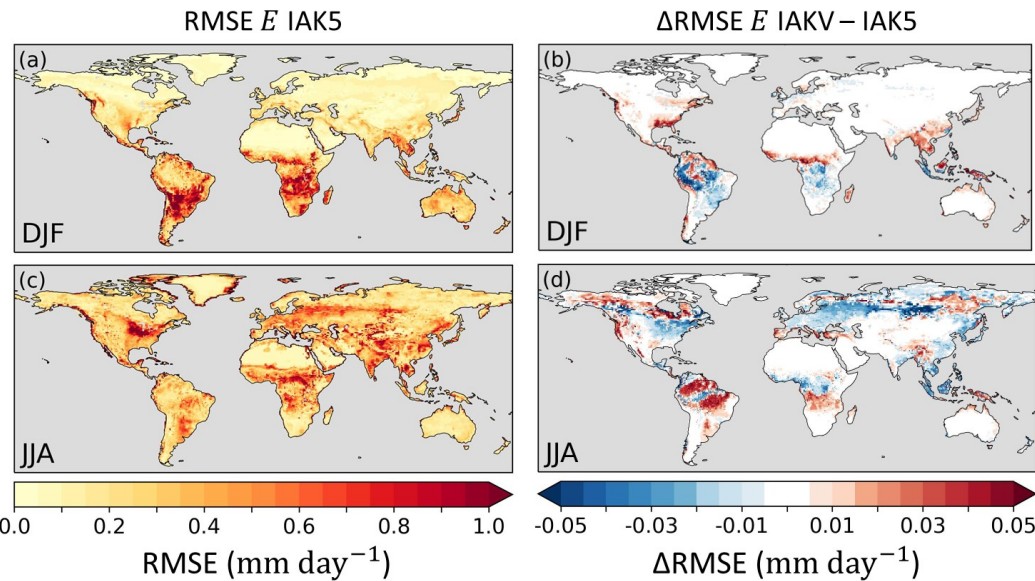

**Figure 13.** Root mean squared error (RMSE) of model seasonal $E$ in experiment IAK5 with respect to DOLCEv3 $E$ for DJF (a) and JJA (c), with red indicating a larger RMSE. The difference between RMSE in IAK5 and IAKV (IAKV−IAK5) for DJF (b) and JJA (d) with blue (red) indicating a reduced (increased) RMSE, and white colors indicate small and/or insignificant ΔRMSE. See Table 1 for details of the experiments. Similar figures for monthly values and all the seasons are presented in Fig. S10-11.

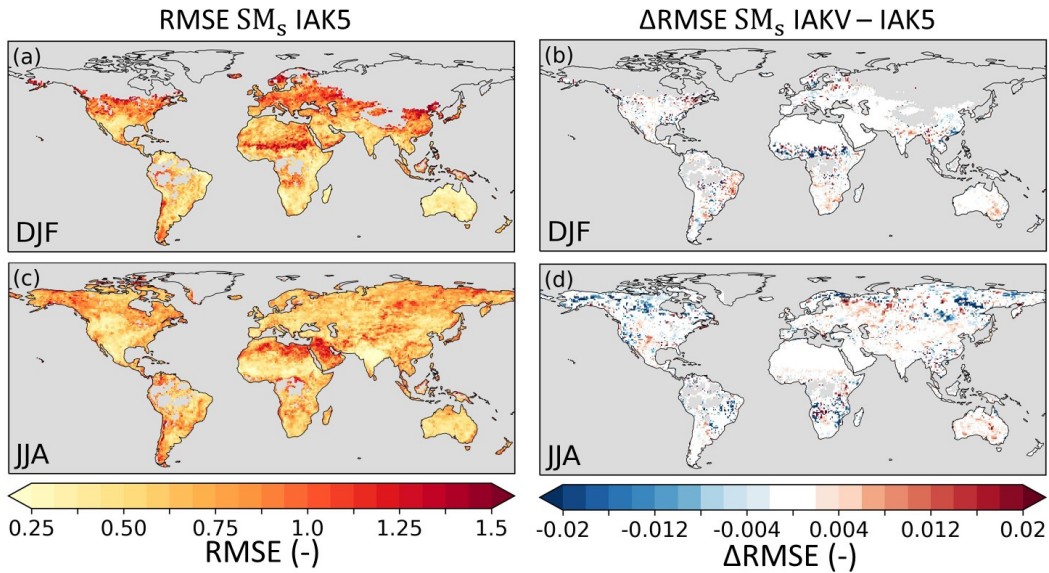

**Figure 14.** Root mean squared error (RMSE) of model standardized inter-annual seasonal anomaly $SM_s$ in experiment IAK5 with respect to ESA-CCI SM for DJF (a) and JJA (c). The difference between RMSE in IAK5 and IAKV (IAKV−IAK5) for DJF (b) and JJA (d) with blue (red) indicating a reduced (increased) RMSE, white colors indicate small and/or insignificant ΔRMSE, and grey indicates no data points. See Table 1 for details of the experiments. Similar figures for monthly values and all the seasons are presented in Fig. S12-13.

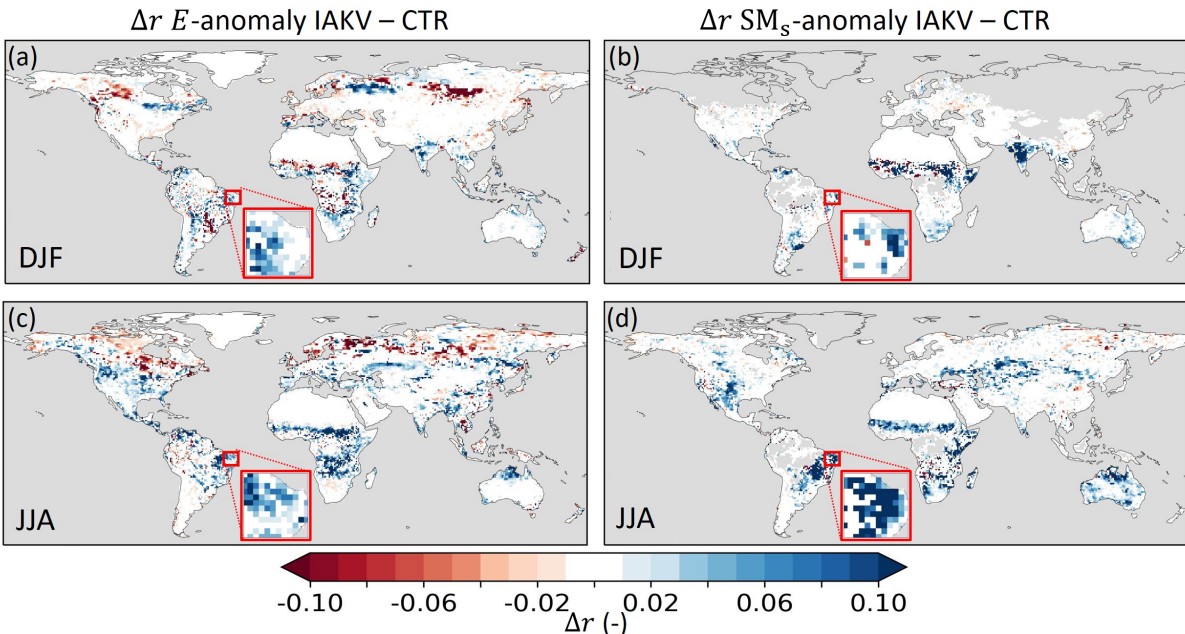

**Figure 15.** Pearson correlation coefficient difference ($\Delta r$) between experiment IAK5 and IAKV (IAKV−IAK5) for (a,c) seasonal anomaly total evaporation ($E$) with respect to DOLCEv3 evaporation for DJF and JJA and (b,d) seasonal anomaly near-surface soil moisture ($SM_s$) with respect to ESA-CCI SM for DJF and JJA. Blue (red) indicates an increased (reduced) correlation in IAKV compared to IAK5, white colors indicate small and/or insignificant $\Delta r$, and grey indicates no data points. The red box is highlighted in Fig. 16. See Table 1 for details of the experiments. Similar figures for all the seasons and monthly anomalies are presented in Fig. S17-19.

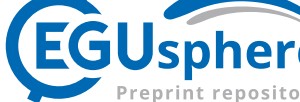

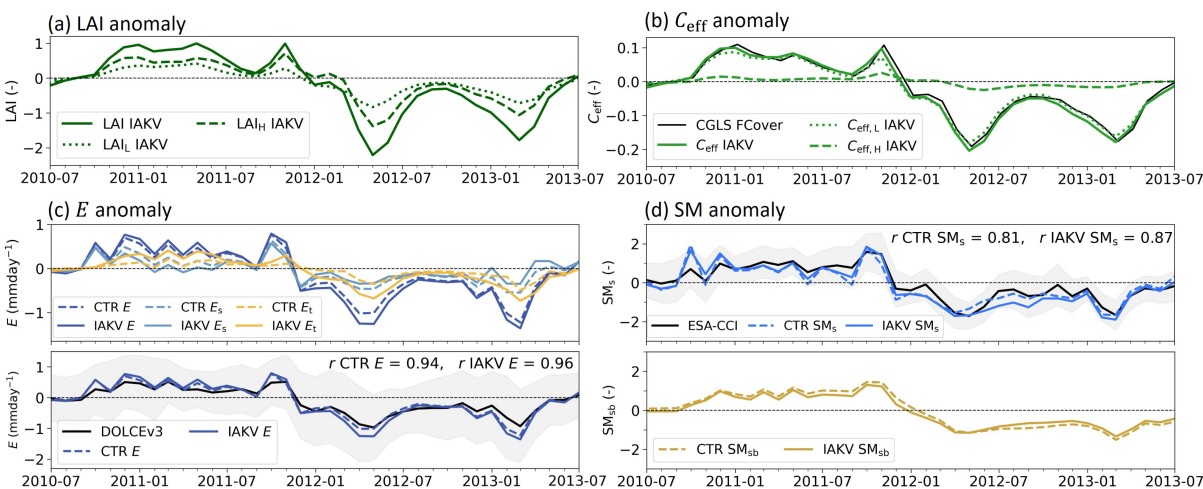

**Figure 16.** Timeseries of the North-Eastern Brazil case highlighted in Fig. 15 for (a) LAI anomalies, (b) Effective vegetation cover ($C_{\text{eff}}$) anomalies with in black the CGLS FCover data as a reference, (c) Evaporation anomalies with $E$ total evaporation, $E_{\text{t}}$ transpiration, $E_{\text{s}}$ soil evaporation and $E_{\text{i}}$ interception evaporation, and in black DOLCEv3 $E$ as a reference, and (d) Soil moisture standardized anomalies with $\text{SM}_{\text{s}}$ near-surface soil moisture and $\text{SM}_{\text{sb}}$ subsurface soil moisture, and in black ESA-CCI SM as a reference. Dashed lines in (c) and (d) represent experiment CTR and solid lines IAKV. The shading in (c) and (d) represents the uncertainty associated with the reference data. For this case $T_{\text{L}}$=Short grass, $T_{\text{H}}$=Deciduous broadleaf trees, $A_{\text{L}}$=0.84 and $A_{\text{H}}$=0.16.

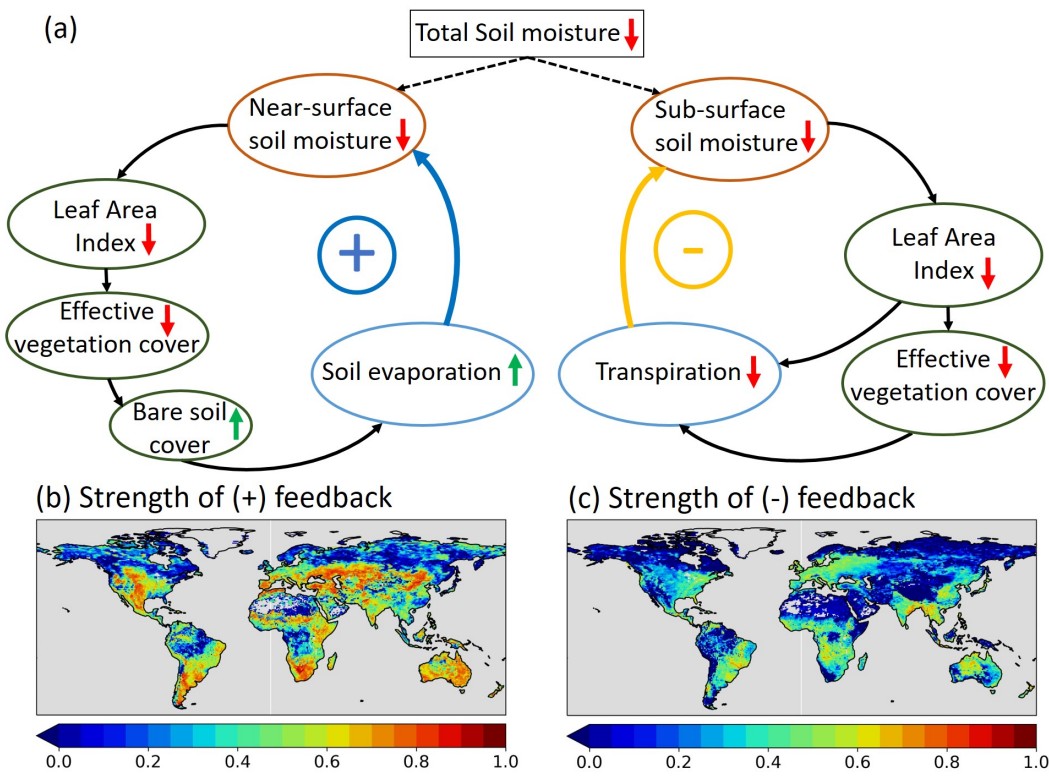

**Figure 17.** (a) Processes contributing to the anomaly vegetation-soil moisture feedback mechanisms as activated with the improved vegetation variability in IAKV compared to CTR. Upward (downward) arrows indicate positive (negative) change in the involved variables. Positive (blue) arrows indicate positive feedback and negative (yellow) arrows indicate negative feedback. +/- refer to the resulting positive/negative feedback loop relative to the sign of the change of the involved variables. The strength of the feedbacks (b,c) is quantified as the absolute correlation between anomaly $\Delta E_s$ and $\Delta SM_s$ (b), and $\Delta E_t$ and $\Delta SM_{sb}$ (c), with $\Delta$ representing the difference between CTR and IAKV (IAKV$-$CTR).