# Peer review of "Inter-annual land cover and vegetation variability based on remote sensing data in the HTESSEL land surface model: implementation and effects on simulated water dynamics"

_EGUsphere, 2023_

## Referee Comment (RC1)

This study evaluates the impact of integrating satellite-derived land cover and vegetation characteristics into the HTESSEL land surface model. By incorporating inter-annually varying land cover, leaf area index, and effective vegetation cover parameterization, the model's representation of evaporation and soil moisture is significantly improved, particularly in regions with changing land cover and during dry seasons. It is an interesting study, making good use of the satellite products available - it is very thorough with a lot of figures and quantitative analysis. Nevertheless, I have a few comments that must be addressed before the manuscript can be published.

**Introduction**

The introduction needs to be expanded. The authors mention that the LSMs do not adequately represent the variability of vegetation, but which models and what do they do instead? The authors also talk about how satellite data has been used to derive ancillary maps, improve parameterisation and evaluate models but don't really talk about how these data are used in data assimilation, which is similar to what is done here. There are a number of studies assimilating LAI (e.g., direct insertion, Kalman filter etc) as well as other satellite products to constrain phenology. I think the study not only benefit from an expanded literature review but also discussing the results in context of other studies. Few examples below:

Albergel, C., Calvet, J.-C., Mahfouf, J.-F., Rüdiger, C., Barbu, A. L., Lafont, S., et al. (2010). Monitoring of water and carbon fluxes using a land data assimilation system: A case study for southwestern France. *Hydrol. Earth Syst. Sci. Discuss.* 14, 1109–1124. doi: 10.5194/hess-14-1109-2010

Kumar, S. V., Mocko, D. M., Wang, S., Peters-Lidard, C. D., & Borak, J. (2019). Assimilation of remotely sensed leaf area index into the Noah-MP land surface model: Impacts on water and carbon fluxes and states over the continental United States. *Journal of Hydrometeorology*, *20*(7), 1359-1377.

MacBean, N., Maignan, F., Peylin, P., Bacour, C., Bréon, F.-M., and Ciais, P.: Using satellite data to improve the leaf phenology of a global terrestrial biosphere model, Biogeosciences, 12, 7185-7208, doi: 10.5194/bg-12-7185-2015

Rahman, A., Zhang, X., Houser, P., Sauer, T., & Maggioni, V. (2022). Global Assimilation of Remotely Sensed Leaf Area Index: The Impact of Updating More State Variables Within a Land Surface Model. *Frontiers in Water*, *3*, 200.

Rahman, A., Zhang, X., Xue, Y., Houser, P., Sauer, T., Kumar, S., et al. (2020). A synthetic experiment to investigate the potential of assimilating LAI through direct insertion in a land surface model. *J. Hydrol. X* 9:100063. doi: 10.1016/j.hydroa.2020.100063

**About ESA-CCI SM**

Note that the ESA-CCI SM combined product uses the GLDAS-Noah model to rescale the different retrievals prior to merging. In theory, this preserves the dynamics and trends of the SSM retrievals but imposes on the combined product the absolute values and dynamic range of GLDAS-Noah (Lui et al., 2012). However, there are also some cases where the dynamics are

also impacted (e.g., Raoult et al., 2022). As such, the authors need to be more mindful when discussing the product in the text:

L231: It is not just the difference in representative soil layers which is an issue, but the construction of the product itself. The merging process uses the climatology and soil depth (10cm) of the GLDAS-Noah model, changing the absolute values.

L414: This needs to be rephrased since it is not strictly true. Although the ESA CCI SM combined product is only made up of remote sensing retrievals, the fact that it does use a land-surface model means that the end product contains information inherited from GLDAS-Noah. Calling it the "most trustworthy" is quite strong. It is a very good global product but deserves more caveating. In fact, I think more text about its limitations is needed in this paragraph.

Liu, Y. Y., Dorigo, W. A., Parinussa, R. M., de Jeu, R. A., Wagner, W., McCabe, M. F., ... & Van Dijk, A. I. J. M. (2012). Trend-preserving blending of passive and active microwave soil moisture retrievals. *Remote Sensing of Environment*, *123*, 280-297.

Raoult, N., Ruscica, R. C., Salvia, M. M., & Sörensson, A. A. (2022). Soil Moisture Drydown Detection Is Hindered by Model-Based Rescaling. *IEEE Geoscience and Remote Sensing Letters*, *19*, 1-5.

**About *k***
More discussion about *k* would be interesting. I realise the values are listed in Figure 10, but maybe a table of the different values for each vegetation in the supplementary materials could be referenced on L180.

L276: Do these values of *k* make sense? Do we expect it to be lower for high vegetation than low vegetation? If is why?

**General**
Figures 2-4 and their discussion belong in the results selection. For example, Figure 4 is first introduced on L161 but with no supporting analysis. Maybe it could be moved to when it is discussed later in the text.

L149: this is not shown anywhere, should Fig 2c be changed to show this? Maybe a stacked bar chart to show the different contributions of vegetation type replacing each vegetation?

Figure 5 would benefit from an extra panel for DOLCEv3 E since it discussed in Section 3.1

**Minor**

I would personally avoid using the word "observations" when referring to retrievals but acknowledge that "observations" is widely used.

The "jet" colour scheme is also no longer recommended for figures

Section 2: add punctuation to the end of equations 10, 11
Section 2.1: rename since doesn't include evaluation sets which are still EO products

L61 : add citations
L80: I believe Earth System Model could be capitalised
L110: I believe it should be 7-189
L113: formatting issue
L212: please put the version number of ESA-CCI SM product used
L215: remind the reader what the model resolution is
L233: what models? Do you mean experiments?
L435: not sure that "obviously" is very sciency

---

## Author Comment (AC1)

*This study evaluates the impact of integrating satellite-derived land cover and vegetation characteristics into the HTESSEL land surface model. By incorporating inter-annually varying land cover, leaf area index, and effective vegetation cover parameterization, the model's representation of evaporation and soil moisture is significantly improved, particularly in regions with changing land cover and during dry seasons. It is an interesting study, making good use of the satellite products available - it is very thorough with a lot of figures and quantitative analysis. Nevertheless, I have a few comments that must be addressed before the manuscript can be published.*

We would like to thank the referee for the comments. We appreciate the time and effort taken to read our manuscript in detail and to provide us the very useful and interesting thoughts on our research. We will take the comments into account when revising the manuscript.

We have separated the different comments (shown in *italic*) and have written our replies below. Text in the original manuscript is shown in '*italic*' and revised text in '***bold***'. Unless differently stated, line numbers mentioned in our reply refer to the original manuscript version.

*Introduction*

*Comment 1.1*

*The introduction needs to be expanded. The authors mention that the LSMs do not adequately represent the variability of vegetation, but which models and what do they do instead? The authors also talk about how satellite data has been used to derive ancillary maps, improve parameterisation and evaluate models but don't really talk about how these data are used in data assimilation, which is similar to what is done here. There are a number of studies assimilating LAI (e.g., direct insertion, Kalman filter etc) as well as other satellite products to constrain phenology. I think the study not only benefit from an expanded literature review but also discussing the results in context of other studies. Few examples below:*

*Albergel, C., Calvet, J.-C., Mahfouf, J.-F., Rüdiger, C., Barbu, A. L., Lafont, S., et al. (2010). Monitoring of water and carbon fluxes using a land data assimilation system: A case study for southwestern France. Hydrol. Earth Syst. Sci. Discuss. 14, 1109–1124. doi:10.5194/hess-14-1109-2010*

*Kumar, S. V., Mocko, D. M., Wang, S., Peters-Lidard, C. D., & Borak, J. (2019). Assimilation of remotely sensed leaf area index into the Noah-MP land surface model: Impacts on water and carbon fluxes and states over the continental United States. Journal of Hydrometeorology, 20(7), 1359-1377.*

*MacBean, N., Maignan, F., Peylin, P., Bacour, C., Bréon, F.-M., and Ciais, P.: Using satellite data to improve the leaf phenology of a global terrestrial biosphere model, Biogeosciences, 12, 7185-7208, doi: 10.5194/bg-12-7185-2015*

*Rahman, A., Zhang, X., Houser, P., Sauer, T., & Maggioni, V. (2022). Global Assimilation of Remotely Sensed Leaf Area Index: The Impact of Updating More State Variables Within a Land Surface Model. Frontiers in Water, 3, 200.*

*Rahman, A., Zhang, X., Xue, Y., Houser, P., Sauer, T., Kumar, S., et al. (2020). A synthetic experiment to investigate the potential of assimilating LAI through direct insertion in a land surface model. J. Hydrol. X 9:100063. doi: 10.1016/j.hydroa.2020.100063*

We expanded the introduction with (the suggested, and other) literature on data assimilation of vegetation properties in other land surface models than HTESSEL.

We modified line 31-36 as follows:

*"**To improve the representation of land surface-atmosphere dynamics, satellite remote sensing data based products have been widely used in LSMs. Global satellite derived maps of land cover and albedo have been directly used as boundary conditions (Faroux et al., 2013; Alessandri et al., 2017; Boussetta et al., 2021). In addition, Leaf Area Index (LAI) derived from satellite remote sensing has been assimilated in several LSMs for different spatial scales, generally leading to improved water, energy and carbon fluxes (Kumar et al., 2019; Ling et al., 2019; Rahman et al., 2020, 2022). Albergel et al. (2017, 2018) also combined LAI assimilation with the assimilation of remote sensing based surface soil moisture in the LSM called ISBA (Interactions between Soil, Biosphere and Atmosphere). This resulted in reduced errors of modeled soil moisture, evaporation, river discharges, and gross primary production with respect to observations.** Furthermore, satellite products have been used to improve model parameterizations of for example **leaf phenology**, surface roughness, soil characteristics, and subsurface water storage (Lo et al., 2010; Trigo et al., 2015; **MacBean et al., 2015**; Yang et al., 2016; Orth et al., 2017). Moreover, LSMs have been evaluated …"*

Additionally, we included a more detailed discussion of our results in relation to other studies in Sect. 4.1 L349:

*"**Recent studies also applied data assimilation methods to integrate satellite-based LAI in LSMs. For example, Rahman et al. (2022) found improved anomaly correlations of transpiration in many areas when integrating satellite based LAI in the LSM called Noah-MP (Noah Multi-Parameterization), with largest effects in the regions where E and $SM_s$ anomaly correlations consistently improved in our results (Fig. 9). However, this study also found limited sensitivity of model surface and root zone soil moisture when only LAI assimilation was applied (Rahman et al., 2022). Similarly, Albergel et al. (2017) concluded that LAI assimilation only affected deeper SM. In contrast, our results showed considerable changes of near-surface soil moisture when integrating CGLS LAI, which can be explained by the interplay between LAI, effective vegetation cover, soil evaporation and near-surface soil moisture schematized in Fig. 17, which apparently differs from the interplay in Noah-MP (Rahman et al., 2020, 2022) and ISBA (Albergel et al., 2017).**"*

Additional references:

Albergel, C., Munier, S., Jennifer Leroux, D., Dewaele, H., Fairbairn, D., Lavinia Barbu, A., Gelati, E., Dorigo, W., Faroux, S., Meurey, C., Le Moigne, P., Decharme, B., Mahfouf, J. F., and Calvet, J. C.: Sequential assimilation of satellite-derived vegetation and soil moisture products using SURFEX-v8.0: LDAS-Monde assessment over the Euro-Mediterranean area, Geoscientific Model Development, 10, 3889–3912, https://doi.org/10.5194/gmd-10-3889-2017, 2017

Albergel, C., Munier, S., Bocher, A., Bonan, B., Zheng, Y., Draper, C., Leroux, D. J., and Calvet, J.-C.: LDAS-Monde Sequential Assimilation of Satellite Derived Observations Applied to the Contiguous US: An ERA-5 Driven Reanalysis of the Land Surface Variables, Remote Sensing, 10, https://doi.org/10.3390/rs10101627, 2018

Ling, X. L., Fu, C. B., Guo, W. D., and Yang, Z. L.: Assimilation of Remotely Sensed LAI Into CLM4CN Using DART, Journal of Advances in Modeling Earth Systems, 11, 2768–2786, https://doi.org/10.1029/2019MS001634, 2019

*About ESA-CCI SM*

*Note that the ESA-CCI SM combined product uses the GLDAS-Noah model to rescale the different retrievals prior to merging. In theory, this preserves the dynamics and trends of the SSM retrievals but imposes on the combined product the absolute values and dynamic range of GLDAS-Noah (Lui et al., 2012). However, there are also some cases where the dynamics are also impacted (e.g., Raoult et al., 2022). As such, the authors need to be more mindful when discussing the product in the text:*

**Comment 1.2**

*L231: It is not just the difference in representative soil layers which is an issue, but the construction of the product itself. The merging process uses the climatology and soil depth (10cm) of the GLDAS-Noah model, changing the absolute values.*

We integrated the information of this comment in L212 as follows:

*"Model near-surface soil moisture ($SM_s$) (0-7 cm) was compared to the **combined active-passive** ESA-CCI soil moisture product (ESA-CCI SM **v06.1**), **which is generated from satellite-based active and passive microwave products that are combined using the absolute values and dynamic range of the modeled soil moisture of the top 10 cm soil layer from the Global Land Data Assimilation System (GLDAS)-Noah LSM (Liu et al., 2012;** Dorigo et al., 2017; Gruber et al., 2019).*

and in L229-L231 as follows:

*Model $SM_s$ and reference ESA-CCI SM cannot be compared directly in absolute terms due to the different representative soil layers and **the imposed dynamic range from the GLDAS-Noah model (Liu et al., 2012), and so potentially** resulting in different temporal variability (Sect. 2.5.2).*

Liu, Y. Y., Dorigo, W. A., Parinussa, R. M., de Jeu, R. A., Wagner, W., McCabe, M. F., ... & Van Dijk, A. I. J. M. (2012). Trend-preserving blending of passive and active microwave soil moisture retrievals. *Remote Sensing of Environment*, *123*, 280-297.

**Comment 1.3**

*L414: This needs to be rephrased since it is not strictly true. Although the ESA CCI SM combined product is only made up of remote sensing retrievals, the fact that it does use a land-surface model means that the end product contains information inherited from GLDAS-Noah. Calling it the "most trustworthy" is quite strong. It is a very good global product but deserves more caveating. In fact, I think more text about its limitations is needed in this paragraph.*

We included the additional information on the ESA-CCI SM generation using GLDAS-Noah model in L212 and L231 (see comment 1.2). Furthermore, we rephrased L411-L414 as follows:

*"Evaluation of the modeled near-surface soil moisture was limited by missing data due to dense forests or snow cover, and the lack of information of the representative soil depth. **While the ESA-CCI combined active/passive SM product was generated using the absolute values and the dynamic range of GLDAS-Noah soil moisture, preserving the dynamics and trends of the original retrievals (Liu et al., 2012), it is important to note that during drydowns the soil moisture dynamics can also be impacted to some extent, as highlighted by Raoult et al. (2022).** However, we still find the ESA-CCI SM **the best suited globally available reference data for our study,** because it is a direct product of remote sensing observations, without **directly blending land surface model dynamics** as done for DOLCEv3."*

**About *k***

*More discussion about k would be interesting. I realise the values are listed in Figure 10, but maybe a table of the different values for each vegetation in the supplementary materials could be referenced on L180.*

To clarify the k-results, we included a table in the supplementary materials (Table S3) and a reference to this table in L277 and the caption of Fig. 10.

*L276: Do these values of k make sense? Do we expect it to be lower for high vegetation than low vegetation? If is why?*

Yes, the values make sense are in line with our expectations (Chen, 2021). The value of *k* represents the amount of vegetation clumping, indicating how much the leaf structure deviates from a random distribution. Low vegetation has more leaves side by side, and, therefore, a larger *k*. For high vegetation, leaves are found more on top of each other, and are, therefore, more clumped and have lower k (Chen, 2005).

We will add the following lines after L276:

**"These findings are in line with our expectations, as leaf organization of low vegetation is more regular (larger k) than leaf organization of high vegetation, where leaves are found more on top of each other (smaller k) (Chen, 2005; Chen 2021)."**

Chen, J. M., Menges, C. H., & Leblanc, S. G. (2005). Global mapping of foliage clumping index using multi-angular satellite data. Remote Sensing of Environment, 97(4), 447-457.

Chen, B., Lu, X., Wang, S., Chen, J. M., Liu, Y., Fang, H., ... & Wang, X. (2021). Evaluation of clumping effects on the estimation of global terrestrial evapotranspiration. Remote Sensing, 13(20), 4075.

**General**

*Figures 2-4 and their discussion belong in the results selection. For example, Figure 4 is first introduced on L161 but with no supporting analysis. Maybe it could be moved to when it is discussed later in the text.*

We agree that Figures 2-4 could have also fitted in the results section. However, we specifically aimed for a clear distinction between preliminary work (i.e., data preparation), and our main contributions (i.e., effective cover parameterization and model evaluation). Therefore, we strongly prefer to keep the structure of the method and results as it is.

To clarify the structure, we will add the following line in L161:

*"Figure 4 shows the LAI inter-annual variability as integrated here in HTESSEL, quantified with the standard deviation. **The effects of this added variability are presented in Sect. 3.2.**"*

L149: this is not shown anywhere, should Fig 2c be changed to show this? Maybe a stacked bar chart to show the different contributions of vegetation type replacing each vegetation?

Figure 2c visualizes these findings, with hatched bars showing low/high vegetation types replacing other low/high vegetation types. For example, crops and short grass have a relatively large hatched bar, compared to the other low vegetation types, indicating crops and short grass were often replacing other low vegetation types.

To clarify this case, we will rephrase L149 as follows:

*".. The figure shows that crops and short grass were relatively often replacing other low vegetation types (relatively large hatched bars), while evergreen needleleaf (EN) and deciduous broadleaf (DB) trees were relatively often replaced by other high vegetation types (relatively large plain bars)."*

Moreover we will add which vegetation types are 'low types' and which are 'high types' in Figure 2c.

*Comment 1.8*

Figure 5 would benefit from an extra panel for DOLCEv3 E since it discussed in Section 3.1

A figure showing the differences of mean E in the experiments CTR and IALC with respect to DOLCEv3 E is now included in the supplement (Figure S1), with reference to this figure in the caption of Fig. 5 and Sect. 3.1 (L272 in the revised manuscript).

**Minor**

*Comment 1.9 -* *I would personally avoid using the word "observations" when referring to retrievals but acknowledge that "observations" is widely used.*

We agree with this comment. We re-read the text and we will change the terms 'observations' in the following lines:

L402: change observations to reference data.

L426: change observations to reference data.

L435: change observations to retrievals.

Title: change satellite observations to remote sensing data (see also comment 2.1).

*Comment 1.10*

*The "jet" colour scheme is also no longer recommended for figures*

We will change the colour schemes.

*Comment 1.11 -* *Section 2: add punctuation to the end of equations 10, 11*

We will change this.

*Comment 1.12 -* *Section 2.1: rename since doesn't include evaluation sets which are still EO products*

We will change the name to '**Land cover and vegetation data**'.

*Comment 1.13 -* *L61 : add citations*

We will add citations.

**_Comment 1.14 -_** _L80: I believe Earth System Model could be capitalized_

We will change this according to the suggestion.

**_Comment 1.15 -_** _L100: I believe it should be 7-189_

We discovered a mistake in line 98 where '100' should be '72'. The fourth layer has a depth of 189, but the total depth of the four layers is 289 cm (layer 1 + 2 + 3 = 100 cm). To avoid this confusion we will clarify this description in lines 98-100 as follows:

_"The subsurface in HTESSEL consists of 4 soil layers **with thicknesses of 7, 21, 72 and 189 cm**. In this study we differentiate between near-surface soil moisture ($SM_s$) in the top layer (0-7 cm), and the subsurface soil moisture ($SM_{sb}$) in the three deeper layers (7-289 cm)."_

**_Comment 1.16 -_** _L113: formatting issue_

We will change this.

**_Comment 1.17-_** _L212: please put the version number of ESA-CCI SM product used_

We will include the version number in the text (v06.1).

**_Comment 1.18 -_** _L215: remind the reader what the model resolution is_

We will change this.

**_Comment 1.19 -_** _L233: what models? Do you mean experiments?_

We meant experiments, so we will change this accordingly.

**_Comment 1.20 -_** _L435: not sure that "obviously" is very sciency_

Agree, we will remove the word 'obviously'.

---

## Author Comment (AC2)

**Referee 2**

*This paper investigates the effects of integrating inter-annually varying land cover and vegetation characteristics, derived from satellite observations, on modeled evaporation and soil moisture. The study uses the Hydrology Tiled ECMWF Scheme for Surface Exchanges over Land (HTESSEL) Land Surface Model (LSM) and data from various sources, including ESA-CCI land cover maps, Copernicus Global Land Services (CGLS) data on Leaf Area Index (LAI), and Fraction of green vegetation Cover (FCover) from CGLS. The paper concludes that integrating interannually varying land cover and vegetation significantly improves the representation of evaporation and soil moisture, highlighting the importance of capturing vegetation variability for accurate modeling of land surface-atmosphere interactions. The findings have important implications for refining land surface models and improving the accuracy of climate change predictions. Overall, this paper is well-written and scientifically sound. It provides valuable insights into the future of LSM modeling in the context of pervasive global change. However, several concerns need to be adequately addressed prior to publication.*

We would like to thank the referee for the comments. We appreciate the time and effort taken to read our manuscript in detail and to provide us the very useful and interesting thoughts on our research. We will take the comments into account when revising the manuscript.

In the review comments, the provided line numbers did not always match the comment. In these cases, we looked at the line numbers that matched the comment, based on our own interpretation of the comment.

We have separated the different comments (shown in *italic*) and have written our replies below. Text in the original manuscript is shown in '*italic*' and revised text in '**bold**'. Unless differently stated, line numbers mentioned in our reply refer to the original manuscript version.

**Comment 2.1**

*The title of the paper does not effectively summarize the main research content and findings. While representing inter-annual land cover and vegetation variability based on remote sensing data is a part of the paper, another part involves studying the improvement of model simulation ability for E and SM after introducing remote sensing data. Therefore, I recommend that the author modify the title accordingly.*

We understand this point and see that the current title does not represent the full scope of the paper. Therefore we will modify the title: *"Representing inter-annual land cover and vegetation variability based on satellite observations in the HTESSEL land surface model"*

to:

**"Inter-annual land cover and vegetation variability based on remote sensing data in the HTESSEL land surface model: implementation and effects on simulated water dynamics"**

*The Introduction section could be further improved. The importance of the research in this paper was not well explained. For example, Line 58: The authors may need to justify the statement "most previous LSM studies aimed at improving the temporally fixed boundary condition of land cover and the monthly seasonal cycle of LAI, while not exploring the effects of inter-annual variations of LC and LAI". Numerous studies have attempted to simulate Leaf Area Index (LAI) using Land Surface Models (LSM). These models can also consider the dynamics of vegetation cover, although uncertainties cannot be ignored. Also, the statement "Moreover, previous studies have generally used one spatially fixed relationship between effective vegetation cover and LAI, ..." requires further explanation and justification.*

We acknowledge that information on other studies on LAI representation in LSMs other than HTESSEL is missing. Therefore, we will further expand the introduction with information on studies integrating satellite-based vegetation properties in LSMs other than HTESSEL (also see comment 1.1). Moreover, we will clarify the research gap that leads to our objective in L58-61 as follows:

*"The research gap that we identified is that most previous LSM studies **using HTESSEL** aimed at improving the temporally fixed boundary condition of land cover and the monthly seasonal cycle of LAI, while not exploring the effects of inter-annual variations of LC and LAI. Moreover, **these** studies have generally used one spatially fixed relationship between effective vegetation cover and LAI, while there is considerable evidence that this relationship is vegetation type dependent (Chen et al., 2005; Ryu et al., 2010; Zhang et al., 2014)."*

**Comment 2.3 -** *Line 77: Why didn't the authors use the AVHRR LAI data directly instead of using a combined dataset from AVHRR and CGLS LAI?*

We acknowledge that we could have also used the AVHRR dataset for LAI for the entire period. However, CGLS LAI is based on the newest sensors, and, therefore, we expect it to be more representative. Moreover, the CGLS products are also of interest because its follow-up data that uses the same processing algorithm is provided operationally in near-real-time. In addition, the 1km resolution of CGLS, which is higher than the 0.05degree resolution of AVHRR, allowed us to better isolate vegetation types for the effective vegetation cover parameterization (see also comment 2.8).

**Comment 2.4 -** *Line 84: Could the authors please provide a brief explanation of the improvements made in this study compared to the previous one? This would help to better understand the novelty of the current study.*

A brief summary of the model improvements made in this study is provided in the introduction L62-67. We intentionally split section 2.2 with the current model representations, and section 2.3 with the model improvements made in this study. To clarify the structure we will make the following changes in the subtitles in section 2:

Section 2.2 from '*Model description*' to **'*Relevant model components for water cycle representation*'.**

Section 2.3: From '*The implemented vegetation variability'* to **'*Model developments*'**

Section 2.3.1: From *'Land cover variability'* to '**The implemented land cover variability'**

Section 2.3.2: From *'Leaf Area Index variability'* to '**The implemented Leaf Area Index variability'**

Section 2.3.2: From *'Vegetation specific effective vegetation cover parameterization'* to '***The implemented vegetation specific effective vegetation cover parameterization***

In addition we will clarify the structure of our methods section by adding the following texts before section 2.1:

***"This section describes how we integrated temporal and spatial variations of land cover and vegetation characteristics in HTESSEL. In Sect. 2.1 we describe the land cover and vegetation data used, in Sect. 2.2 we describe the model characteristics with relevance to water dynamics, and in Sect. 2.3 the model developments performed in this study are reported. Finally, the model experiments and model evaluation are described in Sect. 2.4 and in Sect. 2.5 respectively."***

In addition we will clarify the reference to the model improvements described in section 2.3 in L85 as follows: *"This section describes …. **Section 2.3 describes the adaptations of these model components made in this study."***

**Comment 2.5 -** *Line 100: Why 289 cm? Should it be 189 cm?*

We discovered a mistake in line 98 where '100' should be '72'. The fourth layer has a depth of 189, but the total depth of the four layers is 289 cm (layer 1 + 2 + 3 = 100 cm). To avoid this confusion we will clarify this description in lines 98-100 as follows (see also answer to comment 1.15):

*"The subsurface in HTESSEL consists of 4 soil layers **with thicknesses of 7, 21, 72 and 189 cm**. In this study we differentiate between near-surface soil moisture ($SM_s$) in the top layer (0-7 cm), and the subsurface soil moisture ($SM_{sb}$) in the three deeper layers (7-289 cm)."*

**Comment 2.6 -** *Section 2.2.2 mainly describes how LAI affects RC and W1m, rather than the representation of LAI itself. The same issue applies to Sections 2.2.1 and 2.2.3.*

Section 2.2.1-2.2.3 describe the representation of LC, LAI, and effective vegetation cover in the current HTESSEL version as part of the EC-Earth3 ESM, and the role of these representations in the modelled water cycle. To clarify this we will make the adaptions from comment 2.4, and we will further elaborate on the intention of these sections in L85 as follows:

***"This section describes the relevant model representations of land cover (2.2.1), leaf area index (2.2.2), and effective vegetation cover (2.2.3) in the current HTESSEL version as part of the EC-Earth3 ESM, and the role of these representations in the modelled water cycle. Section 2.3 describes the adaptations of these model components made in this study."***

**Comment 2.7-** *Line 116: Did the authors take the effects of rising CO2 on rc into account?*

No we did not take this into account. To clarify this, we will modify L104-109 as follows:

*"The LAI controls the canopy resistance $r_c$ of the high and low vegetation tiles through the following linear relation:*

$$r_c = r_{s,min} LAI\ f_1(R_s) f_2(D_a) f_3(SM)\ (1)$$

with $r_{s,min}$ the **prescribed** vegetation specific minimum canopy resistance, **that does not change in time, and** $f_1(R_s), f_2(D_a)$ **and** $f_3(SM)$ functions describing the dependencies on shortwave radiation ($R_s$), atmospheric water vapor deficit ($D_a$), and weighted average soil moisture based on the root distribution over the four soil layers (SM), respectively. **The effects of CO2 changes on $r_c$ are not explicitly taken into account in present study.**"

**Comment 2.8 -** *Line 185: The spatial resolution here is approximately 75x75 km. Why was the LAI and LC data interpolated to 1x1 km?*

The LAI and LC data were interpolated to 1x1km for the here developed parameterization of effective vegetation cover. For this parameterization, we aimed to isolate vegetation types (as much as possible) to be able to get the vegetation specific values for k. This isolation of vegetation types was the most representative when using high resolution (i.e. 1x1km) instead of the model resolution (~75x75km), because vegetation type mixing is minimized at higher resolution. For direct model implementation, LAI and LC were used at the model resolution.

We will clarify this in the following lines:

L172: *"To discriminate vegetation types, the grid cells where each vegetation type maximizes its cover fraction based on the ESA-CCI LC developed in Boussetta and Balsamo (2021) were selected for each year. For each set of grid cells corresponding to each vegetation type, the FCover and LAI 10-daily, 1km data for 1999-2019 were extracted.* **Here we used a 1x1km resolution for LAI, FCover and LC in order to obtain the most representative discrimination of vegetation types, and to minimize vegetation mixing within each resolved grid cell.**"

**Comment 2.9 -** *Section 4.1: The section title does not fit in the discussion section. Additionally, I noticed that the text below does not only summarize the results.*

In Section 4.1 we had the intention to combine and synthesize the different results and to elaborate on the implications of our findings. Therefore we will modify the section title from *"Synthesis of results"* to "**Synthesis and implications".**

**Comment 2.10 -** *Line 389: It is misleading to say "fixed atmospheric forcing" here.*

We will change line 389 as follows:

*"Our model experiments were performed in an offline mode with **prescribed** atmospheric forcing,…"*

**Comment 2.11 -** *Section 4.2: While it is commendable to acknowledge these limitations, I suggest that the authors provide a more detailed discussion. For example, they could explore the differences in the representation of land cover and vegetation variability between the models used in this study and the ERA5 LSM, and discuss the potential consequences for the comparative analysis.*

Exploring the differences of vegetation and land cover variability between our models and ERA5 would be an interesting and valuable follow up. However, we believe that exploring this in the discussion of this manuscript would need additional analyses that are beyond the scope of present paper.

**Comment 2.12 -** *Line 435: Many efforts have been made to model global vegetation dynamics, although notable uncertainties still exist. Furthermore, the term 'vegetation evolution' may not be appropriate.*

We will modify L435 as follows:

*"Overall, our results emphasize the need for representing spatial and temporal vegetation variability in LSMs used for climate reanalyses and near-term climate predictions. In climate predictions, we cannot rely on satellite retrievals, and, therefore, the development and validation of dynamical or statistical models able to reliably predict* **vegetation dynamics, from leaf to ecosystem scales, remains** *an important challenge for the future in the land surface modelling community."*

---

## Author Response (AR2)

**Referee #1**

General comments

I applaud the authors' hard work to include all reviewers' remarks. I am satisfied that the authors have adequately addressed my comments, especially by expanding the introduction with more examples and caveating the ESA-CCI SM product.

*We would like to thank the referee again for the time and effort to read our review response and the revised manuscript. We greatly appreciate the recognition of our hard work.*

Minor considerations

1.1  Maybe adding "...totalling a depth of 289cm." to L113 would make it even clearer.

*We will include this sentence to clarify the soil layer depths of the model.*

1.2  I think my confusion around Fig 2 came from forgetting that low vegetation and high vegetation were treated separately, so low vegetation could not replace high vegetation and vice versa. I appreciate the addition of the brackets, but perhaps two separate panels would be more apparent or a reminder in the caption.

*We will include a reminder in the caption of Fig. 2 to clarify that high and low vegetation are treated separately in the model.*

**Referee #2**

I'm happy to see the revised manuscript that addressed my previous comments. I feel that it is much improved and therefore I do not require further improvements.

*We would like to thank the referee again for the time and effort to read our review response and the revised manuscript. We greatly appreciate the recognition of our hard work.*